# Translational Dynamics of Bridled Kites: A Reduced-Order Model in the Course Reference Frame

Oriol Cayon, Vince van Deursen, and Roland Schmehl

Faculty of Aerospace Engineering, Delft University of Technology, 2629 HS Delft, the Netherlands

Correspondence: Oriol Cayon (o.cayon@tudelft.nl)

#### Abstract.


The design and control of airborne wind energy systems requires fast, validated reduced-order models. Because aerodynamic identification of soft, bridled kites is challenging, models that minimise the number of parameters to be identified can be particularly valuable. This paper presents a reduced-order model for the translational dynamics of bridled kites, consisting of a wing supported by multiple bridle lines. The kite is modelled as a point mass in a spherical reference frame aligned with the instantaneous tangential flight direction, referred to as the course reference frame. The angle of attack follows geometrically from a constant angle between the wing chord and the bridle line system, under the assumption that the wing instantaneously aligns with the pull direction, i.e., the rotational dynamics are neglected. The formulation retains gravitational and inertial terms introduced by the curvilinear reference frame and applies a quasi-steady condition of zero path-aligned acceleration, modelling the motion as a sequence of quasi-steady (trimmed) states that relate the trim speed and angle of attack. Model validation is based on public flight datasets from two different soft-wing kites and on dynamic simulations that cover higher wing loadings. Results show that for low wing loadings typical of soft kites, the quasi-steady approximation reproduces the dynamic trajectories with less than 1% deviation in mean reel-out power. For higher loadings and hard-wing kites, inertia introduces substantial phase lag and amplitude damping, causing power deviations of up to 14%. Overall, the proposed model provides a computationally efficient framework for analysing the translational dynamics of bridled kites. The formulation is well-suited to trajectory optimisation, parametric studies, and control design in airborne wind energy systems.

# 1 Introduction

Kites have a long history of use, ranging from recreational and cultural applications to military reconnaissance and atmospheric research (Schmidt and Anderson, 2013). Their first serious applications in engineering emerged in the early 19<sup>th</sup> century, particularly in the field of meteorology, where tethered kites were employed to carry instruments at altitude for atmospheric measurements. In all these early applications, the kite was designed to remain in static equilibrium, generating a lifting force to compensate for its weight and that of the payload.

https://doi.org/10.5194/wes-2025-205

Preprint. Discussion started: 17 October 2025 © Author(s) 2025. CC BY 4.0 License.




It was not until the 1970s that the idea of dynamically flying a kite in crosswind manoeuvres began to emerge. This innovation, eventually popularised through the development of kitesurfing, revealed a key insight: when flown crosswind, a kite can reach speeds several times greater than the ambient wind speed. This leads to a significant increase in aerodynamic forces and thus energy potential.

In the aftermath of the 1970s energy crisis, which stimulated a global search for alternative renewable energy sources, American engineer Miles Loyd recognised the potential of crosswind flight and proposed the use of crosswind-flying kites for generating electricity (Loyd, 1980). In his seminal paper, he derived the fundamental equations governing crosswind flight and provided an initial estimate of the power potential of tethered wings for wind energy generation. His analysis demonstrated that, under idealised conditions, airborne wind energy could extract significantly more power than conventional wind turbines of the same size, highlighting the promise of the technology. However, the theory relied on highly simplified assumptions and neglected several physical effects that were later shown to significantly limit the achievable power in practice (Diehl, 2013).

Since Loyd's original proposal, and particularly in the past two decades, research into airborne wind energy has expanded rapidly. A wide range of modelling approaches have been developed to describe the flight dynamics of tethered wings, spanning from low-fidelity point-mass models to high-fidelity simulations incorporating detailed structural and aerodynamic representations (Vermillion et al., 2021).

At the higher end of this spectrum, the kite is often modelled as a six-degree-of-freedom rigid body, and the tether is discretised as a lumped mass–spring–damper chain to capture its dynamic behaviour (Fechner et al., 2015; Eijkelhof and Schmehl, 2022). These models provide detailed insight into coupled control dynamics but require increased computational resources and large parameter sets (De Schutter et al., 2022). Moreover, the dynamics become less transparent, and intuitive relations between key variables are harder to extract.

Simpler models, by contrast, typically represent the kite as a point mass and the tether as a straight line in quasi-static equilibrium. In many of these formulations, the kite is assumed to fly at a constant lift-to-drag ratio, with aerodynamic forces aligned to the apparent wind direction (Schmehl et al., 2013; Vlugt et al., 2019; Schelbergen and Schmehl, 2020; Fechner and Schmehl, 2013; Ranneberg et al., 2018).

A common assumption in these simplified models is that the motion of the kite can be described as quasi-steady. However, the definition of quasi-steadiness varies across the literature. In some formulations, the inertial forces are assumed to be negligible compared to aerodynamic forces and are therefore omitted entirely (Vlugt et al., 2019; Schelbergen and Schmehl, 2020). In others, only longitudinal and radial accelerations are neglected, while remaining accelerations are accounted for (Schmehl et al., 2013). As a result, there remains a degree of ambiguity in how quasi-steady flight is modelled and interpreted.

This paper introduces a reduced-order formulation of the equations of motion for bridled kites, i.e. systems constrained by a bridle line system. The formulation is developed in a spherical reference frame aligned with the course direction of the kite, which provides a clearer and more intuitive expression of the relevant kinematic quantities. Within this frame, the quasi-steady condition emerges naturally as an implicit property of the system.

Commercial prototypes of bridled kites can be grouped into three main categories: (i) soft kites used in kitesurfing and ground-steered power-generating systems (e.g., Beyond the Sea, SP80, Kitenergy), (ii) soft kites with a suspended control

unit or onboard control surfaces (e.g., Airseas, Kitepower, SkySails Power, Toyota), and (iii) semi-rigid kites with ground-based steering (e.g., Enerkíte). Figure 1 illustrates these prototypes. The present work primarily targets soft-wing kites, such as leading-edge inflatable designs, for which the identification of aerodynamic forces and moments is particularly challenging due to structural flexibility and unconventional aerodynamic shapes (Sánchez-Arriaga et al., 2017).

Figure 1. Beyond the Sea, SP80, Kitenergy, Airseas, Kitepower, SkySails Power, Toyota, and Enerkite (from left top to bottom right).

The remainder of this paper is organised as follows. Section 2 describes the actuation mechanisms of bridled kites and the assumptions underlying the point-mass model. Section 3 derives the equations of motion in a spherical course-aligned frame, and Sect. 4 introduces the quasi-steady simplification that follows from these equations. Section 5 compares the model against experimental data, while Sect. 6 examines quasi-steady flight behaviour and assesses the effect of increasing wing mass on the validity of the assumption. Finally, Sect. 7 summarises the key findings and their implications for optimisation and performance studies.

#### 2 Actuation mechanisms and model assumptions

The control of bridled kites is typically achieved through the adjustment of bridle line geometry, either symmetrically, to adjust the pulling force and flight speed, or asymmetrically, to steer the kite. These actuation strategies directly affect the orientation and magnitude of the aerodynamic force and thus the flight path of the kite. In this section, we describe the two main mechanisms and discuss their implications for the assumptions that are used to build up a point mass model of a kite.

# 2.1 Longitudinal Static Stability and Trim Condition

A key design requirement for bridled kites is longitudinal static stability (Breukels, 2011; Terink et al., 2011), meaning that small deviations in angle of attack generate restoring moments that return the kite to its equilibrium orientation. Such stability ensures the existence of a unique trim angle of attack at which the net moment about the bridle point (B) vanishes, leading to a statically stable equilibrium. In this state, the resultant force at the kite's wing aligns with the resultant force at the bridle point. For a massless kite, the wing force reduces to the aerodynamic force applied at the centre of pressure  $\mathbf{x}_{cp}$ , as illustrated in Figure 2. When weight is included, the equilibrium is preserved by a shift in the trim angle of attack, ensuring that the combined aerodynamic and gravitational forces still align with the bridle resultant. Numerical simulations of symmetric kites in virtual wind tunnels provide evidence of longitudinal static stability, consistently exhibiting convergence towards a unique trim angle of attack at which the net moment about the bridle point vanishes (Poland and Schmehl, 2024b; Cayon et al., 2023; Thedens and Schmehl, 2023), whilst experimental data confirms that the kite's angle of attack remains relatively constant during flight (Oehler and Schmehl, 2019; Schelbergen, 2024; Cayon et al., 2025).

Figure 2. Sideview of symmetric actuation for a schematic massless kite. The depower angle  $\theta_d$  is defined positive in the counter-clockwise direction. Here,  $\mathbf{F}_b$  denotes the resultant force at the bridle/tether attachment point on the kite, including any loads transmitted by a kite control unit (KCU), if present.

The tow angle  $\lambda_b$  is defined as the angle between the front bridle line direction and the straight line from the bridle attachment point to the CP. In the current work, it is assumed that variation of the centre of pressure near the trim point is negligible, implying that  $\lambda_b$  remains approximately constant for a given bridle geometry. This assumption is supported by experimental observations, which show that for leading-edge inflatable (LEI) kites, the force distribution between front and rear bridle lines

Figure 3. Tow angle  $\lambda_b$  as a function of wing angle of attack  $\alpha_w$  for the TU Delft V3 kite. Shaded regions represent the expected range for the reel-in and reel-out phases, based on experimental measurements.

remains approximately constant for a fixed depower setting (Oehler et al., 2018; Oehler and Schmehl, 2019). To substantiate this assumption, Figure 3 plots the tow angle  $\lambda_b$  against the wing angle of attack  $\alpha_w$  for the TU Delft V3 kite, computed with the Vortex Step Method (VSM), a validated lifting-line aerodynamic model (Cayon et al., 2023; Poland et al., 2025); shaded bands indicate the measured reel-out (powered) and reel-in (depowered) ranges (Cayon et al., 2025). Across both regimes,  $\lambda_b$  varies by less than 1°. Thus, for a given bridle geometry, the approximation of a constant geometric angle  $\theta_b$  is justified in the present formulation. Nevertheless, this assumption needs to be re-evaluated for each specific kite design.

Figure 3 also shows an apparent discontinuity near the zero-lift angle of attack, where the centre of pressure is mathematically undefined. As this angle is approached, the centre of pressure shifts aft, producing a nose-down moment that can lead to a front stall. This behaviour is not captured in the present formulation, as it is assumed that the kite is controlled to operate outside this regime.

From Fig. 2, the wing angle of attack  $\alpha_w$  can be related to the bridle angle of attack  $\alpha_b$  by a constant geometric pitch angle  $\theta_b$ ,

$$\alpha_{\mathbf{w}} = \alpha_b - \theta_b,\tag{1}$$

where  $\theta_b$  can be expressed geometrically as a function of the tow angle  $\lambda_b$  and the depower angle  $\theta_d$  alone,

$$\theta_{\rm b} = \theta_{\rm d} + \lambda_{\rm b}. \tag{2}$$

https://doi.org/10.5194/wes-2025-205 Preprint. Discussion started: 17 October 2025 © Author(s) 2025. CC BY 4.0 License.

130

135

# 105 2.2 Symmetric actuation to adjust aerodynamic loading

The preceding discussion implies that the trim state of a bridled kite is uniquely determined by the bridle geometry and the aerodynamic properties of the wing. Therefore, modifying the mean trim angle during operation requires a change in the bridle configuration. In practical systems, this is achieved through symmetric actuation of the bridle lines, which alters both the tow angle and the depower angle.

Depowering the kite corresponds to increasing the depower angle  $\theta_d$ , thereby reducing the trim angle of attack at which the kite operates (see Figure 2). This reduction in angle of attack decreases both the lift-to-drag ratio and the resultant aerodynamic force, leading to lower tangential flight speeds and reduced tether tension. The detailed aerodynamic consequences of depowering will be derived in the following sections.

# 2.3 Asymmetric actuation to steer the kite in turns

While symmetric actuation is used to control the trim angle and flight speed, asymmetric actuation is employed to generate turning manoeuvres by inducing lateral forces and moments. To initiate a turn, a force must be generated perpendicular to both the tether direction and the kite's instantaneous velocity vector. This is achieved by rotating the aerodynamic lift vector towards the centre of the turn. For rigid or semi-rigid wings, this is typically accomplished by physically rolling the wing with respect to the tether axis (Candade, 2023). In contrast, soft kites often achieve this through a combination of body roll and asymmetric deformation (Brown, 1993; Breukels, 2011; Paulig et al., 2013; Bosch et al., 2013; Cayon et al., 2023; Poland and Schmehl, 2023).

Most bridled kites are designed to be directionally stable, i.e. they generate a restoring yawing moment in response to a sideslip (Hur, 2005; Belloc, 2015; Poland et al., 2025). Unlike free-flying aircraft, a wing geometry that would appear unstable about its centre of mass can still be stable once tethered, provided the bridle point is positioned appropriately. In arched kites this typically requires placing the bridle further forward, which ensures that the aerodynamic force distribution produces a restoring yawing moment about the bridle point in response to a sideslip.

In asymmetrically deformed kites, the steering input increases the angle of attack on the inner side of the wing relative to the outer side, generating both a side force and a roll moment. This asymmetry also produces an initial yawing moment that starts the turn. As the kite moves laterally, a sideslip develops. For directionally stable kites, the resulting sideslip produces a yawing moment that maintains the turn. For LEI kites, sideslip angles up to about 5° have been observed for Kitepower's V9 (Cayon et al., 2025). By contrast, in purely roll-driven steering, the roll induces a sideslip angle, which then generates a yawing moment via directional stability.

As the kite turns, the outer wing tip experiences a higher apparent velocity and lower effective angle of attack, while the inner wing tip experiences the opposite (Erhard and Strauch, 2013). This differential shortens the moment arm and produces a yaw-damping effect that resists further rotation, leading to the observed near-linear relation between steering input and yaw rate (Erhard and Strauch, 2013; Fagiano et al., 2013):

$$\dot{\psi} = k u_s v_a. \tag{3}$$

The good agreement with this simplified turn-rate law indicates a quasi-steady yaw response, where the yawing moment equilibrates rapidly and the yaw rate scales proportionally with the product of apparent wind speed and steering input.

# 140 3 Dynamic model formulation

The preceding section outlined the aerodynamic behaviours and actuation mechanisms that govern bridled kites, highlighting the existence of a unique trim condition and the ability to modify the flight state through symmetric and asymmetric inputs. We assume that longitudinal and directional stability drive the wing rapidly towards equilibrium, and therefore, the kite can be approximated as a point mass whose orientation is fixed relative to the force resultant at the bridle point. In this simplified representation, the tow angle  $\lambda_b$  is assumed constant for a given bridle configuration, and the kite is assumed to remain aligned with the apparent wind during controlled flight. The kite's motion is most naturally expressed in a spherical coordinate system centred at the ground station, with components parallel and transverse to the (straight) tether.

# 3.1 Reference frame

The motion of the kite is described using the course reference frame (C-frame), illustrated in Fig. 4, which provides a natural decomposition of the velocity into radial and tangential components. The C-frame origin is located at the ground station, with the unit vectors  $\mathbf{e}_{\chi}$ ,  $\mathbf{e}_{n}$ , and  $\mathbf{e}_{r}$  corresponding to the course, normal, and radial directions, respectively.

The tangential plane, denoted as  $\tau$ , contains  $\mathbf{e}_{\chi}$  and  $\mathbf{e}_{n}$ , and is perpendicular to  $\mathbf{e}_{r}$ . The course angle  $\chi$  defines the orientation of  $\mathbf{e}_{\chi}$  within this plane, with  $\chi=0$  corresponding to motion directly towards the zenith (van Deursen, 2024).

A complete description of the additional reference frames and coordinate transformations is provided in Appendix A.

# 155 3.2 Kinematic relationships in the course reference frame

The translational motion of the kite can be described using Newton's second law of motion, which states that the absolute acceleration  $\frac{d^2\mathbf{r}_k}{dt^2}$  of a point k is equal to the sum of all forces acting upon k, divided by its mass m:

$$\frac{d^2\mathbf{r}_k}{dt^2} = \frac{\sum \mathbf{F}_k}{m}.$$
 (4)

When analyzed in a rotating reference frame, additional terms appear in the acceleration, commonly referred to as fictitious or inertial forces. Below, these quantities are derived in the chosen C-frame.

#### 3.2.1 Velocity

The position vector  $\mathbf{r}_k$  of a point k in the course reference frame is given by  $r\mathbf{e}_r$ . Differentiating with respect to time and applying the product rule yields,

$$\frac{d\mathbf{r}_{\mathbf{k}}}{dt} = \frac{dr}{dt}\mathbf{e}_{\mathbf{r}} + r\frac{d\mathbf{e}_{\mathbf{r}}}{dt} = \begin{bmatrix} 0\\0\\v_r \end{bmatrix} + \mathbf{\Omega}_C \times \mathbf{r}_{\mathbf{k}} = \begin{bmatrix} v_{\tau}\\0\\v_r \end{bmatrix},\tag{5}$$

**Figure 4.** Schematic of the reference frames and aerodynamic angles used in the model. The wind reference frame is shown in black, the azimuth–zenith–radial (AZR) reference frame in orange, and the course reference frame in blue.

where  $\Omega_C$  is the angular velocity of the course reference frame with respect to the inertial wind frame. The velocity vector can thus be written compactly as

$$\mathbf{v}_{\mathbf{k}} = \frac{d\mathbf{r}_{\mathbf{k}}}{dt} = v_{\tau}\mathbf{e}_{\chi} + v_{r}\mathbf{e}_{\mathbf{r}}.$$
(6)

# 3.2.2 Angular velocity of the course reference frame

The C-frame, as explained in Sect. 3.1, is obtained through a sequence of three rotations characterized by the rotation parameters  $\phi$ ,  $\beta$  and  $\chi$ . Since angular velocities are additive, the course reference frame's angular velocity vector  $\Omega_C$  is thus expressed as the sum of the individual rotation rates expressed along their respective axes,

$$\Omega_{c} = \dot{\phi} \mathbf{e}_{z} - \dot{\beta} \mathbf{e}_{\phi} - \dot{\chi} \mathbf{e}_{r} = \begin{bmatrix} \dot{\phi} \cos \chi \cos \beta - \dot{\beta} \sin \chi \\ \dot{\phi} \sin \chi \cos \beta + \dot{\beta} \cos \chi \\ \dot{\phi} \sin \beta - \dot{\chi} \end{bmatrix}.$$
(7)

This form, however, is not very convenient since the derivatives of the elevation  $\dot{\beta}$  and azimuth  $\dot{\phi}$  are already dependent on other kinematic quantities, which can be revealed by solving the system of equations of obtained by equating  $\Omega_{\rm C} \times {\bf r}_{\rm k}$  from

175 Eqs. (5) and (7),

$$\Omega_C \times \mathbf{r}_k = r \begin{bmatrix} \dot{\phi} \sin \chi \cos \beta + \dot{\beta} \cos \chi \\ \dot{\beta} \sin \chi - \dot{\phi} \cos \chi \cos \beta \\ 0 \end{bmatrix} = \begin{bmatrix} v_\tau \\ 0 \\ 0 \end{bmatrix},$$
(8)

with the time derivatives of the position angles

$$\dot{\phi} = \frac{v_{\tau} \sin \chi}{r \cos \beta},\tag{9}$$

$$\dot{\beta} = \frac{v_{\tau} \cos \chi}{r}.\tag{10}$$

Using these expressions, the rotation vector  $\Omega_c$  can now be expressed as a function of the tangential and radial speeds  $v_\tau$ ,  $v_r$ , the course angle  $\chi$  and the course angle rate  $\dot{\chi}$ ,

$$\Omega_{C} = \begin{bmatrix}
\frac{v_{\tau}}{r}\sin\chi\cos\chi - \frac{v_{\tau}}{r}\sin\chi\cos\chi \\
\frac{v_{\tau}}{r}\sin^{2}\chi + \frac{v_{\tau}}{r}\cos^{2}\chi \\
\frac{v_{\tau}}{r}\sin\chi\tan\beta - \dot{\chi}
\end{bmatrix} = \begin{bmatrix}
0 \\
\frac{v_{\tau}}{r} \\
\frac{v_{\tau}}{r}\sin\chi\tan\beta - \dot{\chi}
\end{bmatrix}.$$
(11)

# 3.2.3 Acceleration

The acceleration in the C-frame can be obtained by differentiating Eq. (5) with respect to time, applying the product rule once more,

$$\frac{d^2\mathbf{r}_{\mathbf{k}}}{dt^2} = \frac{d^2r}{dt^2}\mathbf{e}_{\mathbf{r}} + 2\frac{dr}{dt}\frac{d\mathbf{e}_{\mathbf{r}}}{dt} + r\frac{d^2\mathbf{e}_{\mathbf{r}}}{dt^2},$$

which can be expanded and rewritten in terms of the rotation velocity  $\Omega_{c}$  and the position vector  $\mathbf{r}_{k}$  as,

$$\frac{d^2 \mathbf{r}_{\mathbf{k}}}{dt^2} = \frac{d^2 \mathbf{r}_{\mathbf{k}}}{dt^2} \bigg|_{R} + 2\mathbf{\Omega}_{C} \times \frac{d\mathbf{r}_{\mathbf{k}}}{dt} \bigg|_{R} + \mathbf{\Omega}_{C} \times (\mathbf{\Omega}_{C} \times \mathbf{r}_{\mathbf{k}}) + \frac{d\mathbf{\Omega}_{C}}{dt} \times \mathbf{r}_{\mathbf{k}}. \tag{12}$$

Equation (12) shows that the absolute acceleration of k in the C-frame is the summation of the relative acceleration  $\left(\frac{d^2\mathbf{r}_k}{dt^2}\Big|_R\right)$ , the Coriolis acceleration  $\left(2\mathbf{\Omega}_C \times \frac{d\mathbf{r}_k}{dt}\Big|_R\right)$ , the centrifugal acceleration  $\left(\mathbf{\Omega}_C \times (\mathbf{\Omega}_C \times \mathbf{r}_k)\right)$ , and the Euler acceleration  $\left(\frac{d\mathbf{\Omega}_C}{dt} \times \mathbf{r}_k\right)$ , with

$$\frac{d^2 \mathbf{r}_k}{dt^2} \Big|_R = \begin{bmatrix} 0\\0\\\dot{v}_r \end{bmatrix}, \tag{13}$$

$$2\mathbf{\Omega}_{C} \times \frac{d\mathbf{r}_{k}}{dt} \Big|_{R} = \begin{bmatrix} 2\frac{v_{\tau}v_{r}}{r} \\ 0 \\ 0 \end{bmatrix}, \tag{14}$$

195

$$\Omega_{C} \times (\Omega_{C} \times \mathbf{r}_{k}) = \begin{bmatrix} 0 \\ \frac{v_{\tau}^{2}}{r} \sin \chi \tan \beta - v_{\tau} \dot{\chi} \\ -\frac{v_{\tau}^{2}}{r} \end{bmatrix},$$
(15)

$$\frac{d\mathbf{\Omega}_{C}}{dt} \times \mathbf{r}_{k} = \begin{bmatrix} \frac{\dot{v}_{\tau}}{r} - \frac{v_{\tau}v_{r}}{r^{2}} \\ 0 \\ \frac{d}{dt} \left( \frac{v_{\tau}}{r} \sin\chi \tan\beta - \dot{\chi} \right) \end{bmatrix} \times \begin{bmatrix} 0 \\ 0 \\ r \end{bmatrix} = \begin{bmatrix} \dot{v}_{\tau} - \frac{v_{\tau}v_{r}}{r} \\ 0 \\ 0 \end{bmatrix}.$$
(16)

Substituting in Eq. (12), results in the absolute acceleration  $\frac{d^2\mathbf{r}_k}{dt^2}$  in terms of the course reference frame state variables,

$$200 \quad \frac{d^2 \mathbf{r}_k}{dt^2} = \begin{bmatrix} \dot{v}_\tau + \frac{v_\tau v_r}{r} \\ \frac{v_\tau^2}{r} \sin\chi \tan\beta - v_\tau \dot{\chi} \\ \dot{v}_r - \frac{v_\tau^2}{r} \end{bmatrix}. \tag{17}$$

#### 3.3 External forces

The external forces acting on the kite are the aerodynamic force  $\mathbf{F}_a$ , the weight of the kite  $\mathbf{F}_g$  and the tether force  $\mathbf{F}_t$ . These forces must be expressed in terms of the C-frame in accordance to the last section.

# 3.3.1 Gravity force

The most straightforward force is the weight of the kite  $\mathbf{F}_{g}$ , which has a constant direction. Using the transformation  $\mathbb{T}_{C \leftarrow W}$  from Eq. (A4),  $\mathbf{F}_{g}$  is expressed in the C-frame,

$$\mathbf{F}_{g} = -mg\mathbf{e}_{z} = -mg \begin{bmatrix} \cos\chi\cos\beta \\ \sin\chi\cos\beta \\ \sin\beta \end{bmatrix}, \tag{18}$$

where m is the kite mass and g is the gravitational acceleration.

# 3.3.2 Aerodynamic Force

The aerodynamic force  $\mathbf{F}_a$  is composed of drag and lift, both defined relative to the apparent wind vector  $\mathbf{v}_a$ . Drag  $\mathbf{D}$  is aligned with  $\mathbf{v}_a$  by definition, while lift  $\mathbf{L}$  is perpendicular to it. Although asymmetric deformation of the kite can generate side forces, these are not explicitly modelled here; instead, their effect is captured through a control-induced aerodynamic roll angle  $\phi_a$ .

The aerodynamic force can thus be written as

$$\mathbf{F}_a = \mathbf{D} + \mathbf{L}.\tag{19}$$

215 Decomposing the apparent wind vector in the C-frame yields

$$\mathbf{v}_{a} = \begin{bmatrix} v_{a,\chi} \\ v_{a,n} \\ v_{a,r} \end{bmatrix} = \begin{bmatrix} v_{w,\chi} - v_{\tau} \\ v_{w,n} \\ v_{w,r} - v_{r} \end{bmatrix}, \tag{20}$$

The drag force is then

$$\mathbf{D} = \frac{1}{2} \rho S C_D(\alpha_{\mathbf{w}}) v_a \begin{bmatrix} v_{a,\chi} \\ v_{a,n} \\ v_{a,r} \end{bmatrix}. \tag{21}$$

Lift is assumed to act in the plane normal to  $\mathbf{v}_a$ , and its direction is determined by the aerodynamic roll angle  $\phi_a$ . This angle accounts for both the control-induced roll (e.g., via asymmetric deformation or physical roll of the wing) and the roll induced by the kite control unit. The lift vector is expressed as:

$$\mathbf{L} = \frac{1}{2}\rho SC_L(\alpha_{\mathbf{w}})v_a^2 \mathbf{e}_L,\tag{22}$$

$$\mathbf{e}_{L} = \frac{1}{v_{a}v_{a,\tau}} \begin{bmatrix} -v_{a}v_{a,n}\sin\phi_{a} - v_{a,\chi}v_{a,r}\cos\phi_{a} \\ v_{a}v_{a,\chi}\sin\phi_{a} - v_{a,n}v_{a,r}\cos\phi_{a} \\ v_{a,\tau}^{2}\cos\phi_{a} \end{bmatrix}, \tag{23}$$

where  $v_{a,\tau} = \sqrt{v_{a,\chi}^2 + v_{a,n}^2}$  is the component of the apparent wind in the tangential plane. The derivation of the lift direction is provided in Appendix D1.

The wing angle of attack  $\alpha_w$  is obtained under the assumptions that the kite remains aligned with the apparent wind and that the pitch angle between the wing chord and the resultant force at the bridle point is constant (see Appendix D2).

The aerodynamic coefficients  $C_L(\alpha_w)$  and  $C_D(\alpha_w)$  are obtained by interpolating aerodynamic polar curves. The sideslip angle is not explicitly modelled, but its effect on the total aerodynamic lift is assumed negligible, based on prior numerical and experimental studies showing only minor degradation at the small sideslip angles observed during flight (Viré et al., 2022; Cayon et al., 2023; Poland et al., 2025).

#### 3.3.3 Tether Force

230

235

A realistic tether can only be loaded axially and therefore deforms due to gravity, aerodynamic drag, and inertial forces. For this simplified model, a straight, inelastic and inertia-free tether is assumed. The effective weight and drag of the tether acting on the kite are obtained from a quasi-static equilibrium by enforcing moment balance at the ground station, which implies that kite tangential accelerations are not included in the tether model. A schematic of the force components is shown in Fig. 5.

The net tether force at the kite is obtained from a moment balance about the ground station, incorporating the effects of tether weight and aerodynamic drag. The drag force is approximated as acting at the kite in the direction of the apparent wind velocity (Vlugt et al., 2019).

Figure 5. Free body diagram of a straight, axially loaded tether in a spherical coordinate frame.

240 This yields the following expressions for the tangential and normal components of the tether force at the kite:

$$F_{t,\tau} = -\frac{1}{2}\rho_t gr\cos\chi\cos\beta + \frac{1}{8}\rho C_{D,c} d_t r v_a (v_{w,\chi} - v_\tau), \tag{24}$$

$$F_{t,n} = -\frac{1}{2}\rho_t g r \sin \chi \cos \beta + \frac{1}{8}\rho C_{D,c} d_t r v_a v_{w,n}. \tag{25}$$

The radial component is:

$$F_{t,r} = -F_{tg} - \rho_t g r \sin \beta + \frac{1}{8} \rho C_{D,c} d_t r v_a (v_{w,r} - v_r). \tag{26}$$

The full derivation is provided in Appendix D3.

# 3.4 Equations of motion

Having defined the absolute acceleration and the external forces in the C-frame, the translational dynamics of a tethered kite follow from Newton's second law,

$$m\frac{d^2\mathbf{r}_k}{dt^2} = \mathbf{F}_{\text{ext}} = \mathbf{F}_a + \mathbf{F}_t + \mathbf{F}_g.$$
 (27)

250 The model is formulated as a system of differential-algebraic equations (DAEs):

$$\dot{\mathbf{x}} = \mathbf{f}(\mathbf{x}, \mathbf{z}, \mathbf{u}) = \begin{bmatrix} v_r & \frac{v_\tau \cos \chi}{r} & \frac{v_\tau \sin \chi}{r \cos \beta} & \dot{\chi} & \dot{v}_r & \dot{v}_\tau \end{bmatrix}, \tag{28}$$

$$\mathbf{0} = \mathbf{g}(\mathbf{x}, \mathbf{z}, \mathbf{u}) = m \frac{d^2 \mathbf{r}_k}{dt^2} - \mathbf{F}_{\text{ext}}.$$
 (29)

Here, x, z, and u denote the differential states, algebraic states, and control inputs, respectively

$$\mathbf{x} = \begin{bmatrix} r & \beta & \phi & \chi & v_r & v_\tau \end{bmatrix},\tag{30}$$

$$\mathbf{z} = \begin{bmatrix} \dot{v}_{\tau} & \dot{\chi} & F_{t,g} \end{bmatrix}, \tag{31}$$

$$\mathbf{u} = \begin{bmatrix} \dot{v}_r & u_s & u_p \end{bmatrix}. \tag{32}$$

In this work,  $u_s$  is the steering input (actuation that primarily sets the aerodynamic roll and thereby the course rate  $\dot{\chi}$ ), and  $u_p$  is the depower input (actuation that changes the geometric pitch  $\theta_b$  and thus affects the angle of attack  $\alpha_w$ ).

# 4 Quasi-steady equilibrium

In the context of crosswind flight, the quasi-steady state is defined as the trimmed condition arising from the instantaneous balance of forces and moments acting on the system. As the kite's orientation relative to the wind changes along its trajectory, the trim condition evolves with its position and motion direction in the wind window.

#### 4.1 Definition and Assumptions

To illustrate the governing balance in its simplest form, we first consider an idealised case in which the kite is positioned at the centre of the wind window ( $\phi = 0$ ,  $\beta = 0$ ), with no tether dynamics included. In this scenario, the tangential acceleration  $\dot{v}_{\tau}$  depends only on the tangential speed  $v_{\tau}$ , motion direction  $\chi$ , reeling speed  $v_{\rm r}$ , and control inputs (depower  $u_{\rm p}$  and steering  $u_{\rm s}$ ). The governing equation reduces to

$$m\dot{v}_{\tau} = \frac{1}{2}\rho S v_a \left[ C_L(\alpha_w) \cos \phi_a(v_w - v_r) - C_D(\alpha_w) v_{\tau} \right] - mg \cos \chi. \tag{33}$$

The same interpretation applies at any position, although the explicit form of the aerodynamic terms is more complex.

Plotting  $\dot{v}_{\tau}$  as a function of  $v_{\tau}$  in Fig. 6 shows that it typically crosses zero at two points. These crossings correspond to candidate quasi-steady equilibria defined by

$$\dot{v}_{\tau} = 0. \tag{34}$$

However, only the equilibrium that satisfies the local stability criterion

$$\frac{\partial \dot{v}_{\tau}}{\partial v_{\tau}} 

290

295

Figure 6. Tangential acceleration  $\dot{v}_{\tau}$  as a function of the tangential speed  $v_{\tau}$  and angle of attack  $\alpha_{\rm w}$ , for different course angles  $\chi$ . Results shown for the Kitepower V9 kite. A negative slope near the equilibrium confirms local stability.

angle of attack. Conversely, during descent ( $\chi=180^{\circ}$ ), gravity assists the motion, allowing the force vector to rotate backward and reducing the required trim angle. Because the trim angle of attack and tangential speed are linked by the aerodynamic equilibrium, a higher  $\alpha_{\rm w}$  corresponds to a lower  $v_{\tau}$ , and vice versa. Consequently, changes in the gravitational component along the course lead to different equilibrium speeds, even under the quasi-steady assumption. The characteristic acceleration and deceleration in flight patterns typically attributed to gravity are thus captured implicitly within the quasi-steady solution, without the need for explicit modelling of dynamic inertial effects.

These observations support an interpretation of the kite dynamics as continuously converging toward a moving quasi-steady state, defined by the instantaneous position, motion direction, and control inputs. When aerodynamic forces dominate and the wing loading (m/S) is sufficiently small, this convergence is rapid enough to approximate the motion as a sequence of quasi-steady states. This assumption is further examined in Sect. 6.3, where dynamic and quasi-steady simulations are compared across a range of wing loadings.

This treatment differs from earlier implementations, where inertial accelerations were sometimes omitted entirely (Vlugt et al., 2013; Schelbergen and Schmehl, 2020), or where tangential and radial accelerations were assumed negligible compared to aerodynamic contributions (Schmehl et al., 2013). In contrast, the present formulation retains the inertial terms and defines the quasi-steady equilibrium through the condition of zero tangential acceleration, corresponding to the trimmed state of the kite.

However, for practical implementation, we also assume the radial acceleration imposed by the winch,  $\dot{v}_r$ , to be negligible. This simplification, adopted in earlier quasi-steady models, is justified by the relatively small winch acceleration (Schmehl et al., 2013).

# 4.2 Quasi-steady equations of motion

Following the definition of quasi-steady equilibrium, the dynamic DAE system in Eq. (28) can be reduced by eliminating the differential states associated with the radial and tangential accelerations,  $\dot{v}_{\rm r}$  and  $\dot{v}_{\tau}$ . The resulting state vectors are

$$\mathbf{x}_{\mathrm{qs}} = \begin{bmatrix} r & \beta & \phi & \chi \end{bmatrix},\tag{36}$$

$$\mathbf{z}_{qs} = \begin{bmatrix} v_{\tau} & \dot{\chi} & F_{tg} \end{bmatrix}, \tag{37}$$

$$\mathbf{u}_{qs} = \begin{bmatrix} v_r & u_s & u_p \end{bmatrix},\tag{38}$$

where  $\mathbf{x}_{qs}$  contains the remaining position and orientation variables,  $\mathbf{z}_{qs}$  the algebraic variables associated with tangential speed, course rate, and tether force, and  $\mathbf{u}_{qs}$  the control inputs. The reduced quasi-steady formulation is thus expressed as a semi-explicit DAE system of index 1:

$$\frac{d\mathbf{x}_{qs}}{dt} = \mathbf{f}(\mathbf{x}_{qs}, \mathbf{z}_{qs}, \mathbf{u}_{qs}), 
\mathbf{0} = \mathbf{g}(\mathbf{x}_{qs}, \mathbf{z}_{qs}, \mathbf{u}_{qs}),$$
(39)

where f describes the reduced differential kinematics and g enforces instantaneous force balance.

The quasi-steady formulation is independent of the time history: at each instant the state is obtained from the algebraic force balance  $\mathbf{g}(\mathbf{x}, \mathbf{z}, \mathbf{u}) = \mathbf{0}$  at the current position and inputs. By contrast, the dynamic formulation is history-dependent and must be solved as an initial-value problem.

# 5 Validation of quasi-steady model



The quasi-steady model is validated using flight data from two kites of different sizes: the TU Delft V3 kite (Poland and Schmehl, 2024a) and the V9 kite from Kitepower (Cayon et al., 2024), with publicly available datasets that enable reproducibility. Key parameters of the two systems are summarised in Table C1. Notably, the V3 system was equipped with a kite control unit (KCU) whose mass was approximately twice that of the wing, which is atypical for properly scaled systems and is expected to influence the dynamics and feasibility of the quasi-steady assumption.

The validation is conducted by imposing the measured flight trajectories as inputs to the quasi-steady model. At each recorded time step, the measured position  $(r, \beta, \phi)$ , course angle  $\chi$  and rate  $\dot{\chi}$ , radial speed  $v_{\rm r}$ , and wind speed  $v_{\rm w}$  are prescribed as inputs. The quasi-steady model is then used to compute the corresponding tangential speed  $v_{\tau}$ , tether force  $F_{\rm tg}$ , and required steering input  $u_{\rm s}$ , which are compared to the measurements.

It is important to note that the wind speed used in the quasi-steady reconstruction differs between the two cases. For the V3 kite, the wind speed was estimated using an extended Kalman filter (EKF) specifically tailored for soft kites (Cayon et al., 2025). In contrast, the V9 case used lidar measurements taken around 200m upwind of the kite and interpolated to the kite height. However, the lidar data is subject to 1-minute temporal averaging, which smooths out short-term fluctuations. Conversely, the EKF reconstruction for the V3 flight may also struggle to resolve rapid wind changes. As a result, even if

https://doi.org/10.5194/wes-2025-205 Preprint. Discussion started: 17 October 2025 © Author(s) 2025. CC BY 4.0 License.





the model perfectly reproduced the underlying physics, discrepancies between the predicted and measured quantities may still arise due to limitations in wind and state estimation.

#### 5.1 Aerodynamic identification

Aerodynamic modelling of flexible kites remains one of the most challenging aspects of kite design. The arched geometry and extensive recirculation zones induced by the unconventional leading-edge inflatable (LEI) airfoils complicate accurate aerodynamic simulation. Recent wind tunnel experiments with the V3 kite have demonstrated that neither CFD simulations nor simplified models based on lifting-line theory can reliably reproduce the aerodynamic behaviour of these kites. In particular, both the magnitude and slope of the drag coefficient are consistently underestimated, suggesting that neither parasitic nor induced drag components are captured adequately by current modelling approaches (Poland et al., 2025). Moreover, these discrepancies do not yet account for structural deformations, which further increase the gap between simulation and reality. Experimental observations have revealed significant deformation of the three-dimensional kite geometry, including bending of the inflatable struts, which directly affects aerodynamic performance. Additional phenomena such as trailing edge flutter and bridle line vibrations also contribute to deviations in aerodynamic characteristics.

Given these complexities, purely simulation-based aerodynamic identification often fails to accurately represent the true behaviour of deformable kites. Consequently, a semi-empirical approach combining both simulation data and experimental measurements is adopted to achieve a more reliable aerodynamic characterisation.

Experimental data obtained during flight tests allow the estimation of the mean lift and drag coefficients corresponding to three representative flight states: (i) powered and straight flight during reel-out, (ii) powered and steered flight during reel-out, and (iii) depowered flight during reel-in. The baseline aerodynamic polars are first computed using the vortex step method, a validated lifting-line-based model (Cayon et al., 2023; Poland et al., 2025), suitable for low aspect ratio and curved geometries, and second-order polynomial fits are applied to both the lift and drag curves. Subsequently, for each of the three representative states, a parasitic drag offset is added to the drag curve such that the corresponding  $C_L$ – $C_D$  polar intersects the experimentally identified coefficients for that state (see Fig. 7).

The lift coefficient is modelled as a second-order polynomial function of the angle of attack

$$C_L(\alpha_{\rm w}) = C_{L,0} + C_{L,1}\alpha_{\rm w} + C_{L,2}\alpha_{\rm w}^2. \tag{40}$$

The drag coefficient incorporates both the baseline drag curve and empirical corrections to account for control-induced effects. It is expressed as

$$C_D(\alpha_{\rm w}, u_p, u_s) = C_{D,0} + C_{D,1}\alpha_{\rm w} + C_{D,2}\alpha_{\rm w}^2 + C_{D,p}u_p + C_{D,s}u_s, \tag{41}$$

where  $u_p$  and  $u_s$  are the depower and steering control inputs, respectively. The terms  $C_{\rm D,p}$  and  $C_{\rm D,s}$  introduce multiplicative corrections to capture the increase in drag associated with depower and steering, while  $C_{\rm D,0}$  accounts for a baseline parasitic drag offset, representing the drag observed in straight powered flight. The polynomials for the TU Delft V3 kite can be found in Appendix C.

Figure 7. Aerodynamic polar diagram showing  $C_L$  versus  $C_D$  for the TUDELFT V3 kite. The baseline curve is obtained from VSM simulations, with a quadratic fit applied. Semi-empirical corrections are introduced to match three experimentally identified flight states.

Finally, the aerodynamic roll angle  $\phi_a$  is empirically characterised as a linear function of the steering input  $u_s$ , based on flight test data

$$\phi_{\mathbf{a}} = k_{\phi,\mathbf{s}} u_{\mathbf{s}}.\tag{42}$$

The resulting modified polars incorporate both the baseline aerodynamic behaviour and empirical corrections derived from flight tests, effectively accounting for the drag contributions of the bridle lines, KCU, and onboard turbine.

Similar correction strategies have been successfully employed in previous quasi-steady kite modelling studies (Vlugt et al., 2019; Schelbergen and Schmehl, 2020), offering a practical compromise between model fidelity and computational tractability.

# 5.2 Comparison with experimental data



With the identified aerodynamic polars, assumed linearly dependent on steering and depower inputs, the quasi-steady model is used to retrace the force, tangential speed, and steering input required to sustain each measured state. This enables a direct comparison between the model predictions and experimental measurements at each point along the flown trajectory, under the assumption of instantaneous aerodynamic equilibrium.

Figure 8 shows segments of two representative flights, comparing measured and estimated tangential speed and tether force, as well as pitch relative to the radial direction and aerodynamic roll, against the corresponding pitch and roll relative to the radial direction calculated by the EKF (Cayon et al., 2025), which employs a discretised tether model rather than the simplified model used here.



**Figure 8.** Validation of the quasi-steady model against flight data from two kite systems. For TU Delft V3 kite (one cycle) and Kitepower V9 kite (two cycles), the figure compares measured and reconstructed tangential speed, tether force, and steering input.

In the reel-out phase, the temporal response of the model is good, with the predicted peaks in tangential speed and tether force aligning closely with those measured. However, the estimated values appear noisier—particularly for the V9 kite—and some peaks are overestimated. The kite pitch with respect to the radial direction agrees well in both trend and magnitude with the estimations obtained from a discretised tether model. In contrast, the roll is compared against the aerodynamic roll, which accounts for both the KCU-induced roll and the kite side-force, and therefore yields higher values during turning manoeuvres than those estimated by the EKF. For the V3 kite, which was equipped with an unusually heavy KCU relative to its size, several states cannot be resolved in a quasi-steady manner—especially on the lower part of the figure-eight, where the kite exits the turn and begins to climb. This explains the gaps observed in the corresponding estimations.

In the reel-in phase, the model reproduces the behaviour of the measured quantities reasonably well, provided that the depower setting is adjusted to yield a lift coefficient consistent with the values inferred from measurements. Under this condition, the magnitude variation of the tether force and tangential speed are in line with the measurements, although small discrepancies remain. Here, both the pitch and the aerodynamic roll align well with the EKF estimations, as the KCU primarily affects the pitch axis during reel-in.

https://doi.org/10.5194/wes-2025-205 Preprint. Discussion started: 17 October 2025

© Author(s) 2025. CC BY 4.0 License.





The transition from reel-in to reel-out remains the most challenging phase for the model. In this regime, the predicted tether force often overshoots the measurements. These deviations arise from two main factors: the kite undergoes dynamic manoeuvres that violate the quasi-steady assumption, and the tether can exhibit significant sag under low tension, followed by a rapid transition to high tension that is not well captured by the simplified straight-line tether model. Together, these effects reduce the model's accuracy during this phase. For the V9 kite, this behaviour appears at the beginning of the reel-out phase—immediately after transition—because of how the masks are defined, but it effectively corresponds to the first turn into reel-out.

The purpose of this comparison is to assess whether the overall behaviour and magnitude are consistent, rather than to achieve a direct point-by-point match, since the state variables used to reconstruct the quasi-steady equilibrium carry noise and inaccuracies. Moreover, the wind is not measured directly at the kite, but either with lidar or estimated through an EKF, both of which introduce additional uncertainties.

# 400 6 Analysis of Quasi-steady and Dynamic Flight Behaviour

This section investigates the behaviour of crosswind flight through a combination of quasi-steady parametric analyses and dynamic simulations. The quasi-steady framework is first used to explore how kite position and reeling strategy influence performance metrics such as tangential speed and power extraction. Subsequently, dynamic simulations are employed to assess the validity of the quasi-steady approximation, highlighting the role of inertia and its impact on flight response.

### 405 6.1 Influence of Kite Position on Quasi-steady Tangential Speed

The position of the kite within the wind window, that is, its azimuth and elevation relative to the wind direction, significantly influences the aerodynamic forces and resulting tangential velocity. As the kite moves away from the centre of the wind window, the component of the wind velocity perpendicular to the wing surface decreases. Consequently, the tangential speed  $v_{\tau}$  required to achieve a quasi-steady equilibrium ( $\dot{v}_{\tau}=0$ ) diminishes, since a lower apparent wind angle relative to the wing reduces the required flight speed to maintain the trim angle of attack.

This behaviour is illustrated in Fig. 9, which shows the non-dimensional tangential speed factor  $\lambda = v_{\tau}/v_w$  as a function of the azimuth, elevation, and course angles, under the assumption of straight flight ( $\dot{\chi}=0$ ). As seen in Fig. 9a, the tangential speed decreases with the elevation of the kite, and as the elevation increases, the dependency on the course angle becomes greater, mostly due to the aerodynamic force getting closer to the weight. The tangential speed is maximal at  $\chi=180^{\circ}$ , where gravity assists the motion, and minimal at  $\chi=0^{\circ}$ , where it opposes it.

A similar dependency is observed with respect to the azimuth angle  $\phi$ , where increasing misalignment reduces the tangential speed. The combined effect of azimuth and course angles shifts the location of maximum tangential speed. As shown in Fig. 9b, for  $\phi = 0^{\circ}$ , the maximum occurs at  $\chi = 180^{\circ}$ , but as  $\phi$  increases, both the maximum and minimum shift toward  $\chi = 270^{\circ}$ , where the kite points more directly into the centre of the wind window. This shift results from the interplay between the wind incidence angle and the tangential projection of gravity, which together influence the equilibrium speed.

Figure 9. Isolines of elevation and azimuth angles as a function of the tangential speed factor  $\lambda$  and the course angle  $\chi$  for  $v_r = 0$ . Results shown for the Kitepower V9 kite.

### 6.2 Optimal Reel-out Strategy


The reeling speed plays a major role in determining both the tether force and the harvested power. It is well-known that, for a simplified crosswind operation at the centre of the wind window and neglecting gravity, the optimal reeling speed is a fixed fraction of the wind speed, originally derived by Loyd (1980) to be  $v_r = \frac{1}{3}v_w$ . Extending this result to arbitrary positions within the wind window—while still neglecting gravity—the optimal reeling factor becomes dependent on both elevation and azimuth, and is given by (Schmehl et al., 2013)

$$v_{\rm r,opt} = \frac{v_{\rm w}}{3}\cos\phi\cos\beta. \tag{43}$$

This positional dependency reflects the reduction in crosswind efficiency with increasing elevation  $\beta$  and off-centre azimuth  $\phi$ . These trends, now accounting for both gravitational effects and course angle, are illustrated in Fig. 10. The results show that maximum reeling speeds are obtained near  $\chi=180^\circ$ , where gravity assists the motion and increases tether force, whereas the optimal reeling factor decreases with increasing elevation or azimuth due to reduced tangential speed and diminished tether loading. From Fig. 10, one can observe a consistent relationship between the optimal reeling speed and the corresponding tether force across different positions. This motivates the analytical derivation—under the assumption of quasi-steady flight and neglecting gravity—of a direct expression linking the two, given the lift and drag coefficients of the wing, which can be obtained with the simplified equations derived in Schmehl et al. (2013),

$$F_t = 2\rho S C_R v_{r,opt}^2 \left[ 1 + \left( \frac{C_L}{C_D} \right)^2 \right]. \tag{44}$$

**Figure 10.** Isolines of elevation and azimuth angles as a function of the instantaneous optimal reeling factor. Results shown for the Kitepower V9 kite.

This relation suggests a practical control strategy in which the winch reeling speed is regulated as a function of the measured tether force (Berra and Fagiano, 2021; Hummel et al., 2024). Since  $F_t$  inherently captures the combined influence of wind speed and kite position, this enables an implicit, adaptive reeling control scheme without the need for precise wind measurements or position-dependent logic. The inclusion of gravitational effects alters the force equilibrium and shifts the aerodynamic trim,

**Figure 11.** Instantaneous optimal reeling speed as a function of the tether tension for different conditions. Results shown for the Kitepower V9 kite.

leading to deviations from the idealised relation in Eq. (44). This displacement is illustrated in Fig. 11, where the optimal reeling speed is plotted as a function of tether tension for multiple positions and orientations the wind window for different wind speeds. Nonetheless, for the wing loadings typical of soft kites, the deviation introduced by gravity remains relatively small, allowing the analytical relation in Eq. (44) to serve as an effective basis for reeling control.

It is important to note that while the analytical expression represents the instantaneous reeling speed that maximises power extraction, it does not necessarily correspond to the optimal reel-out speed over a full pumping cycle. The cycle-averaged optimum depends not only on the reel-out phase but also on the duration, dynamics, and efficiency of the reel-in phase. Consequently, an effective reeling strategy must take into account the full cycle for optimal power generation (Luchsinger, 2013).

# 6.3 Dynamic Response in Crosswind Trajectories

To analyse the system response along prescribed flight paths, the kite motion is parametrised using a scalar path coordinate s(t), which evolves in time. This formulation enables all state variables to be expressed as functions of s and its time derivatives, simplifying the comparison between dynamic and quasi-steady responses. The resulting velocity and acceleration components are derived analytically in terms of s,  $\dot{s}$ , and  $\ddot{s}$ . The current parametrisation defines the kite trajectory on the wind window (elevation and azimuth) independently of its radial position, which makes it possible to specify the reeling strategy separately from the angular path. The complete formulation, including the path speed and kinematic derivatives in spherical coordinates, is provided in Appendix B.

Two representative trajectories are considered for crosswind flight in the W-frame. The first corresponds to a circular path, while the second describes a Lissajous figure-eight. In both cases, the trajectory is parametrised by a shape parameter s and time t, allowing all state variables to be expressed as functions of t, s, its derivatives  $\dot{s}$ ,  $\ddot{s}$ , the steering input  $u_s$ , and the tether force  $F_{tq}$ . A constant reel-out speed is assumed for clarity of comparison between the dynamic responses,

$$r(t) = r_0 + v_r t. (45)$$

The trajectories are defined as:

- Circular path:

460

$$\beta = \beta_c + \frac{\Delta \varphi}{2} \sin(s), \qquad \phi = \phi_c + \frac{\Delta \varphi}{2} \cos(s). \tag{46}$$

465 – Lissajous figure-eight:

$$\beta = \beta_c + \frac{\Delta \beta}{2} \sin(2s), \qquad \phi = \phi_c + \frac{\Delta \phi}{2} \cos(s). \tag{47}$$

where  $\Delta \phi$  and  $\Delta \beta$  denote the horizontal and vertical amplitudes of the patterns, respectively, and  $\phi_c$  and  $\beta_c$  the center position of the pattern.

Figure 12. Parametrized helix and figure of eight patterns.

Two integration schemes are applied. In the dynamic scheme, for each time step, the path acceleration  $\ddot{s}$ , the tether force  $F_{tg}$ , and the steering input  $u_s$  are obtained by solving the force equilibrium given the current state  $(s, \dot{s})$ , and are subsequently integrated to update  $\dot{s}$  and s. This is written as a differential-algebraic equation (DAE) system in semi-explicit form,

$$\frac{d}{dt} \begin{bmatrix} s \\ \dot{s} \end{bmatrix} = \begin{bmatrix} \dot{s} \\ \ddot{s} \end{bmatrix}, 
\mathbf{0} = \mathbf{g} \underbrace{\left( \underbrace{(s, \dot{s})}_{T}, \underbrace{(\ddot{s}, F_{tg}, u_{s})}_{T} \right)} \tag{48}$$

where the algebraic constraint g enforces instantaneous force balance along the prescribed trajectory.

In contrast, the quasi-steady scheme assumes that the tangential acceleration vanishes, i.e.  $\dot{v}_{\tau} = 0$ , such that  $\ddot{s}$  need not be computed. In this case, only the path speed  $\dot{s}$  is obtained from the force equilibrium, and s is advanced using a single integration step. The system reduces to an algebraic-differential formulation:

$$\mathbf{0} = \mathbf{g}\left(\underbrace{(s)}_{x}, \underbrace{(\dot{s}, F_{tg}, u_{s})}_{z}\right) \tag{49}$$

where the algebraic equation determines the steady-state value of  $\dot{s}$  consistent with force equilibrium at each point along the trajectory.

Figures 13 and 14 compare dynamic, quasi-steady and inertia-free simulations of a parameterized trajectory across four representative kite configurations, covering typical mass and aerodynamic characteristics of both soft and rigid wings: two soft kites (TU Delft V3 (Poland and Schmehl, 2024a) and Kitepower V9) and two rigid wings (AP2 (Malz et al., 2019) and a 100 kW MegAWES (optimized)). The soft kite models correspond to systems used as validation in section 5, flying figure-of-eight trajectories with reeling speeds and dimensions representative of their real-world operation. In contrast, the rigid wings are simulated under idealised yet representative flight conditions, following circular trajectories with fixed radius and constant



reeling speed that are optimized based on maximum reel-out power generation. The angle between kite and tether is chosen such that the trim angle of the kite corresponds to The details of the aerodynamic characteristics, kite sizes and weights, and flown trajectories can be found in Appendix C.

Figure 13. Comparison between dynamic and quasi-steady simulations for the TU Delft V3 and Kitepower V9 soft-kite configurations. The two subplots on the left show the kite trajectories in terms of azimuth and elevation, with colour indicating the tangential speed  $v_{\tau}$ . Results are shown for both the dynamic and quasi-steady cases. The four time series on the right represent one full flight loop and display the evolution of tangential speed  $v_{\tau}$ , normalised tether force  $(\overline{F}_{t,g})$ , steering input  $(u_s)$ , and angle of attack  $(\alpha_w)$ . The normalisation of tether force is performed with respect to the mean.

The results in Figure 13 show that for both the V3 and V9 soft kites, the quasi-steady approximation closely matches the dynamic simulation. For the V9 kite, the dynamic, quasi-steady, and inertia-free simulations are nearly indistinguishable in both mean values and time evolution, showing only a slight phase lag of the dynamic response relative to the quasi-steady and inertia-free results. For the V3 kite, the mean values of velocity and tether force remain comparable to those in the quasi-steady simulation (see Table 1), but the temporal evolution exhibits larger deviations, with higher maximum speeds and a more damped tether-force behaviour. The dynamic simulation of the V3 also reveals stronger oscillations in the angle of attack, likely due to an oversized KCU, whereas the oscillations for the V9 remain below 2°. This difference is further reflected in the greater roll of the lift vector observed for the V3. Overall, the quasi-steady approximation reproduces the main dynamic behaviour of

Figure 14. Comparison between dynamic and quasi-steady simulations for the AP2 and MegAWES 100 kW rigid wing systems. The two subplots on the left show the kite trajectories in terms of azimuth and elevation, with colour indicating the tangential speed  $v_{\tau}$ . Results are shown for both the dynamic and quasi-steady cases. The four time series on the right represent one full flight loop and display the evolution of tangential speed  $v_{\tau}$ , normalised tether force  $(\overline{F}_{t,g})$ , aerodynamic roll angle  $(\phi_a)$ , and angle of attack  $(\alpha_w)$ . The normalisation of tether force is performed with respect to the mean.

**Table 1.** Comparison between dynamic and quasi-steady simulations for the four kite configurations analysed. All values are expressed as percentage differences relative to the dynamic simulations, except for the phase shift which is given in degrees.

| Kite           | Wing loading | $\Delta P$ (%) | $\Delta F_{t,\min/\max}$ (%) | $\Delta v_{\tau, \min/\max}$ (%) | $\Delta\Phi_{v_{	au,\min/\max}}$ (°) |
|----------------|--------------|----------------|------------------------------|----------------------------------|--------------------------------------|
| TU Delft V3    | 1.83         | -0.77          | 43.26 / -14.72               | -2.21 / 6.68                     | -9.67 / -9.98                        |
| Kitepower V9   | 2.00         | -0.4           | 3.22 / -2.99                 | -1.26/ 1.67                      | -5.63 / -5.87                        |
| AP2            | 12.27        | -9.38          | -8.95 / -3.74                | -15.18 / 4.60                    | -51.84 / -51.10                      |
| MegAWES 100 kW | 30.00        | -12.00         | -11.85 / -3.25               | -21.88/ 8.33                     | -74.33 / -63.56                      |

both kites with minor discrepancies in phase and amplitude, and with negligible differences in average power—below 1 % for both cases.

https://doi.org/10.5194/wes-2025-205 Preprint. Discussion started: 17 October 2025 © Author(s) 2025. CC BY 4.0 License.







The results for the rigid wings, shown in Figure 14, display more pronounced deviations between the dynamic and quasisteady simulations than observed for the soft kites. In addition to a clear phase lag, a significant amplitude attenuation is evident in the dynamic trajectories. This damping effect is particularly pronounced in the minima of tangential velocity and tether tension. The quasi-steady model, which neglects tangential acceleration  $\dot{v}_{\tau}$ , compensates the weight with a steep increase in angle of attack to maintain equilibrium, resulting in sharp oscillations. In contrast, the dynamic simulations maintain a more gradual evolution of the aerodynamic state, with a smoother variation in angle of attack. This reflects the system's limited capacity to respond instantaneously due to its larger inertia.

The phase lag is substantially greater than for the soft kites, with the maximum tangential speed and tether tension shifted by more than  $70^{\circ}$  in the worst-case scenario (see Table 1). This delay, combined with the reduced amplitude of oscillations observed in the dynamic simulations, leads to significant deviations in the predicted power output: -7.0% for the AP2 wing and -13.9% for the MegAWES 100 kW system. Interestingly, the damping of these oscillations results in a higher overall power estimate in the dynamic model compared to the quasi-steady prediction. This behaviour can be attributed to the kite's inertia, which allows it to ascend without requiring the excessively high angle of attack demanded by the quasi-steady model. As a result, the dynamic system remains closer to an aerodynamically optimal state throughout the trajectory. These findings highlight the growing importance of incorporating dynamic effects at higher wing loadings, where quasi-steady assumptions become increasingly inadequate for accurate performance evaluation and control design.

For the inertia-free assumption, the discrepancies are even more pronounced. The delay in both tangential speed and tether force relative to the dynamic simulation increases further, while the predicted roll input remains unrealistically small throughout the cycle. In this case, the maximum angles of attack are also reduced, since only the weight needs to be balanced and inertial loads are absent.

Despite these discrepancies, a key dynamic behaviour observed in the soft kite simulations persists in the rigid-wing cases: the kite accelerates or decelerates whenever the quasi-steady tangential velocity intersects the dynamic trajectory. This demonstrates that the dynamic state remains attracted to the quasi-steady solution, with the system continuously responding in its direction. While the convergence is not instantaneous, due to increased inertia in the rigid configurations, the dynamic model still reveals a tendency to track the quasi-steady state. This shared behaviour across all kite types supports the interpretation of the quasi-steady solution as a moving target towards which the system naturally evolves. This supports the use of quasi-steady models as predictive tools, provided their limitations are recognised in the context of higher wing loading configurations.

### 7 Conclusions

This work presents a simplified model for the translational dynamics of bridled kites, relevant to airborne wind energy and ship propulsion applications. The model assumes that the kite rapidly achieves a trimmed aerodynamic state due to its low rotational inertia relative to the aerodynamic forces and moments. This justifies a point-mass formulation without enforcing a constant angle of attack, allowing the aerodynamic forces to be resolved based on the instantaneous trim condition. As a result, the model provides a more intuitive understanding of the interplay between angle of attack and kite speed, which underpins the

https://doi.org/10.5194/wes-2025-205 Preprint. Discussion started: 17 October 2025 © Author(s) 2025. CC BY 4.0 License.








physical basis of crosswind flight. Specifically, a more orthogonal wind incidence necessitates a higher flight speed to maintain equilibrium, explaining the structure of the wind window and the high energy potential of crosswind motion.

The model is developed in the course reference frame, a spherical coordinate system aligned with the kite's tangential velocity. This facilitates an intuitive decomposition of velocity and acceleration into radial and tangential components, enabling a clear analysis of inertial effects. Within this framework, the quasi-steady condition is naturally defined as a state of zero tangential acceleration, which corresponds to a continuously adapting trim state. The decomposition also provides physical insight into the inertial forces experienced by the kite as it moves along a spherical path and turns within the constraints imposed by the tether, making it possible to interpret these fictitious forces meaningfully even in a quasi-steady framework.

A key insight from the model is that the kite's weight is the primary factor influencing the trim angle of attack. In the absence of changes to bridle geometry or control input, variations in the gravitational force component along the flight path directly alter the force balance. As the kite moves with the direction of gravity, the trim angle of attack decreases, requiring a higher flight speed to sustain equilibrium—even within a quasi-steady framework.

Validation against experimental data from two different kite systems demonstrates the applicability of the model, particularly in replicating the locations of maximum and minimum tangential speeds. This suggests that soft kites generally operate near a quasi-steady regime during crosswind flight. However, accurate estimation of aerodynamic polar curves remains essential. Due to the complex and deformable nature of soft kites, numerical methods frequently underpredict drag. To address this, empirical corrections were applied to the simulated aerodynamic coefficients based on flight data.

Comparative analyses of quasi-steady and dynamic models for reel-out trajectories reveal the influence of kite inertia. For wing loadings representative of soft kites, the quasi-steady approximation remains valid. However, with increasing mass, deviations become more pronounced, highlighting the limits of the quasi-steady approach for heavier systems.

Despite its strengths, the model exhibits several limitations. Firstly, the model neglects rotational dynamics, assuming that the kite instantaneously reaches equilibrium. Secondly, the tether is modeled as a straight, inertia-free element. While this simplifies computations, it introduces inaccuracies during low-tension manoeuvres, especially when tether sag becomes non-negligible.

Future extensions of this work will focus on trajectory optimisation and path planning, leveraging the computational efficiency of the quasi-steady framework. A more realistic representation of the reeling speed (e.g. dependent on the tether force) should also be incorporated, which can be readily achieved thanks to the independent parametrisation of the tangential plane and radial direction. Moreover, the impact of the simplified tether model should be analysed, as an improved representation may be particularly relevant for simulating low-tether-force scenarios such as during reel-in.

In conclusion, the proposed model offers a computationally efficient yet physically grounded framework for analysing bridled kite dynamics, particularly under crosswind flight. Its scope is primarily soft, bridled kites such as leading-edge inflatable designs, where point-mass modelling provides a practical alternative to high-fidelity rigid-body approaches; for rigid-wing systems, models with explicit aerodynamic moment identification remain more appropriate. The present formulation is devised for optimisation applications and control design of lightweight bridled kites.

Code availability. The code will be made available in the final production version.

# Appendix A: Reference frames and transformations

In addition to the course reference frame described in Sect. 3.1, two additional frames are introduced to define the kite's position and orientation: the wind reference frame (W) and the azimuth-zenith-radial reference frame (AZR).

## 570 Wind reference frame (W)

The W-frame is a Cartesian reference frame with its origin at the ground station  $O_G$ . The  $\mathbf{e}_x$  unit vector aligns with the mean wind direction at a reference height, while  $\mathbf{e}_z$  points vertically upward from the Earth's surface. Effects of the Earth's rotation on the kite's motion are neglected in this frame, treating it as inertial.

# Azimuth-Zenith-Radial reference frame (AZR)

The AZR-frame is a rotating reference frame in which the position of the kite is expressed using spherical coordinates  $(\phi, \beta, r)$ , where  $\phi$  is the azimuth angle,  $\beta$  is the elevation angle, and r is the radial distance.

The elevation angle  $\beta$  is measured between the  $(\mathbf{e}_x, \mathbf{e}_y)$ -plane and  $\mathbf{r}_k$ , while the azimuth angle  $\phi$  is measured between the  $(\mathbf{e}_x, \mathbf{e}_z)$ -plane and  $\mathbf{r}_k$ . The position of the kite is thus given by

$$\mathbf{r}_{\mathbf{k}} = r\mathbf{e}_{\mathbf{r}}.$$
 (A1)

The transformation from the W-frame to the AZR-frame is obtained through two sequential rotations:

$$\mathbb{T}_{AZR\leftarrow W} = \mathbb{R}_{\mathbf{x}'} \left( \frac{\pi}{2} - \beta \right) \mathbb{R}_{\mathbf{z}} \left( \phi + \frac{\pi}{2} \right), \tag{A2}$$

where  $\mathbb{R}_{\mathbf{x}'}(\cdot)$  and  $\mathbb{R}_{\mathbf{z}}(\cdot)$  denote standard rotation matrices.

# Transformation from C-frame to W-frame

The C-frame is obtained by rotating the AZR-frame around the radial direction  $\mathbf{e}_{r}$  by an angle  $\frac{\pi}{2} - \chi$ , aligning  $\mathbf{e}_{\chi}$  with the tangential velocity:

$$\mathbb{T}_{C \leftarrow AZR} = \mathbb{R}_{\mathbf{r}} \left( \frac{\pi}{2} - \chi \right). \tag{A3}$$

The total transformation from the W-frame to the C-frame reads:

$$\mathbb{T}_{C \leftarrow W} = \mathbb{T}_{C \leftarrow AZR} \mathbb{T}_{AZR \leftarrow W}. \tag{A4}$$

# Appendix B: Path parametrization framework

Let  $\mathbf{R}(s)$  be the parametrisation of the position vector  $\mathbf{r}_k$  of a point k, such that

$$\mathbf{r}_k(t) = \mathbf{R}(s(t)). \tag{B1}$$

This implies that simulating the motion of k along a prescribed trajectory reduces to solving for the time-dependent path coordinate s(t). Differentiating with respect to time yields

$$\frac{d\mathbf{r}_k}{dt} = \frac{d\mathbf{R}}{ds}\frac{ds}{dt},\tag{B2}$$

and taking the dot product of both sides with  $d\mathbf{R}/ds$  gives

$$\frac{d\mathbf{r}_k}{dt} \cdot \frac{d\mathbf{R}}{ds} = \frac{ds}{dt} \left\| \frac{d\mathbf{R}}{ds} \right\|^2. \tag{B3}$$

By definition of the dot product, and since  $\frac{d\mathbf{r}_k}{dt}$  and  $\frac{d\mathbf{R}}{ds}$  are aligned,

$$\frac{d\mathbf{r}_k}{dt} \cdot \frac{d\mathbf{R}}{ds} = \left\| \frac{d\mathbf{r}_k}{dt} \right\| \left\| \frac{d\mathbf{R}}{ds} \right\|. \tag{B4}$$

Substituting this into the earlier expression, we obtain the path speed

$$\dot{s} = \frac{v_k}{\left\| \frac{d\mathbf{R}}{ds} \right\|},\tag{B5}$$

where  $v_k = \left\| \frac{d\mathbf{r}_k}{dt} \right\|$  is the magnitude of the kite velocity.

# **B1** Parametrisation in the AZR Frame

Let the spherical coordinates  $(\phi, \beta, r)$  of the kite position be expressed as

$$\phi = \phi(s(t)), \qquad \beta = \beta(s(t)), \qquad r = r(t).$$
 (B6)

The position vector becomes  $\mathbf{R}(s) = r\mathbf{e}_r$ , and its derivative is

$$\frac{d\mathbf{R}}{ds} = \frac{dr}{ds}\mathbf{e}_r + r\frac{d\mathbf{e}_r}{ds}.\tag{B7}$$

where Using the angular velocity of the AZR-frame,

$$\frac{d\mathbf{e}_r}{ds} = \mathbf{\Omega}_{AZR} \times \mathbf{r}, \qquad \mathbf{\Omega}_{AZR} = \frac{d\phi}{ds} \mathbf{e}_z - \frac{d\beta}{ds} \mathbf{e}_\phi, \tag{B8}$$

the derivative becomes

$$\frac{d\mathbf{R}}{ds} = \begin{bmatrix} r \frac{d\phi}{ds} \cos \beta \\ r \frac{d\beta}{ds} \\ \frac{dr}{ds} \end{bmatrix}_{\phi,\beta,r}$$
(B9)

Thus, the norm is

$$\left\| \frac{d\mathbf{R}}{ds} \right\| = \sqrt{\left(\frac{dr}{ds}\right)^2 + r^2 \left(\frac{d\beta}{ds}\right)^2 + r^2 \left(\frac{d\phi}{ds}\right)^2 \cos^2 \beta}.$$
 (B10)

# **B2** Velocity Components

The radial velocity can be written as

$$v_{\rm r} = \frac{dr}{ds}\dot{s}$$
. (B11)

Given  $v_k = \sqrt{v_\tau^2 + v_r^2} = \left\| \frac{d\mathbf{R}}{ds} \right\| \dot{s}$ , we obtain the tangential speed

$$v_{\tau} = \dot{s}r\sqrt{\left(\frac{d\beta}{ds}\right)^2 + \left(\frac{d\phi}{ds}\right)^2 \cos^2\beta}.$$
(B12)

#### **B3** Kinematic Derivatives

From Eqs. (10) and (9), the course angle can be expressed by

$$\tan \chi = \frac{\dot{\phi}\cos\beta}{\dot{\beta}}$$
 (B13)

The course rate is given by the chain rule

$$\dot{\chi} = \frac{d\chi}{ds}\dot{s},\tag{B14}$$

where

$$\frac{d\chi}{ds} = \frac{\frac{d^2\phi}{ds^2} \frac{d\beta}{ds} \cos\beta - \frac{d\phi}{ds} \frac{d^2\beta}{ds^2} \cos\beta - \frac{d\phi}{ds} \left(\frac{d\beta}{ds}\right)^2 \sin\beta}{\left(\frac{d\beta}{ds}\right)^2 + \left(\frac{d\phi}{ds}\right)^2 \cos^2\beta}.$$
(B15)

The radial acceleration becomes

$$\dot{v}_r = \frac{d^2r}{ds^2}\dot{s}^2 + \frac{dr}{ds}\ddot{s},\tag{B16}$$

and the tangential acceleration reads

$$\dot{v}_{\tau} = \left(\dot{s}^2 \frac{dr}{ds} + \ddot{s}r\right) \sqrt{A} + \frac{1}{2} \dot{s}r \frac{\dot{A}}{\sqrt{A}},\tag{B17}$$

with

$$A = \left(\frac{d\beta}{ds}\right)^2 + \left(\frac{d\phi}{ds}\right)^2 \cos^2\beta,$$
 (B18)

$$\dot{A} = 2\dot{s} \left[ \frac{d\beta}{ds} \frac{d^2\beta}{ds^2} + \frac{d\phi}{ds} \frac{d^2\phi}{ds^2} \cos^2\beta - \left( \frac{d\phi}{ds} \right)^2 \frac{d\beta}{ds} \cos\beta \sin\beta \right]. \tag{B19}$$

# Appendix C: System description and path parameters

This appendix serves to describe all the input parameters used in the presented results and simulations.

**Table C1.** Main system parameters of the simulated kites. Angle of attack  $\alpha_w$  in radians.  $\theta_t$  denotes the geometric pitch angle between the wing chord and the tether axis, as imposed by the bridle configuration (see Eq. (2)). Aerodynamic characteristics of the Kitepower V9 kite are not disclosed for confidentiality reasons.

|                                                   | TU Delft V3          | Kitepower V9 | Ampyx AP2           | 100 kW MegAWES      |
|---------------------------------------------------|----------------------|--------------|---------------------|---------------------|
| Wing mass $m_{ m w}$ [kg]                         | 14.2                 | 62           | 36.8                | 444                 |
| KCU mass $m_{ m kcu}$ [kg]                        | 22                   | 31.6         | N/A                 | N/A                 |
| Wing projected area $A [m^2]$                     | 19.75                | 46.85        | 3                   | 15.44               |
| Tether diameter $d_t$ [mm]                        | 10                   | 14           | 2.5                 | 10                  |
| Coeffs $C_L$ $(C_{L,0}, C_{L,1}, C_{L,2})$        | (0.17, 5.69, -10.78) | _            | (0.55, 5.04, -5.27) | (0.3, 6.96, -2.889) |
| Coeffs $C_D$ $(C_{D,0}, C_{D,1}, C_{D,2})$        | (0.14, -0.18, 1.79)  | _            | (0.05, -0.04, 1.10) | (0.01, 0.06, 0.39)  |
| Actuation $C_D$ coeffs $(k_p, k_s)$               | (0.01, 0.04)         | _            | N/A                 | N/A                 |
| Reel-out tether–wing pitch angle $\theta_t$ [deg] | 9                    | _            | _                   | _                   |
| Reel-in tether–wing pitch angle $\theta_t$ [deg]  | 31                   | =            | N/A                 | N/A                 |

# **System characteristics**

The parameters in Table C1 define the aerodynamic and geometric properties of each kite configuration considered. Mass, area, and tether diameter are directly specified, while lift and drag polynomials are expressed as second-order functions of the angle of attack. For the TU Delft V3 and Kitepower V9 kites, additional actuation-dependent drag terms are included. The tether—wing pitch angle  $\theta_t$ , imposed by the bridle geometry, is listed separately for reel-in and reel-out phases when applicable.

# Path characteristics

**Table C2.** Path parameters. Figure-eight requires both azimuth width  $\Delta \psi$  and elevation height  $\Delta \beta$ . Circular paths require only one angular span (set the unused one to N/A).

| Parameter                            | TU Delft V3 | Kitepower V9 | Ampyx AP2 | 100 kW MegAWES |
|--------------------------------------|-------------|--------------|-----------|----------------|
| Path type (circle / fig-8)           | fig-8       | fig-8        | circle    | circle         |
| Azimuth center $\phi_c$ [deg]        | 0           | 0            | 0         | 0              |
| Elevation center $\beta_c$ [deg]     | 32          | 28           | 25        | 25             |
| Azimuth width $\Delta\psi$ [deg]     | 20          | 40           | 14        | 12             |
| Elevation height $\Delta\beta$ [deg] | 10          | 20           | 14        | 12             |
| Initial radius $r_0$ [m]             | 200         | 220          | 400       | 600            |
| Reel-out speed $v_r$ [m/s]           | 1           | 1.5          | 3.6       | 3.14           |

The path definitions in Table C2 specify the spatial loops used in the simulations. Figure-eight trajectories are characterized by both an azimuthal span and an elevation span, while circular trajectories are defined by a single angular extent. The center angles  $(\psi_c, \beta_c)$  determine the mean positioning of the loop relative to the wind direction, and the initial tether length  $r_0$  fixes the loop's radius. The imposed reel-out velocity  $v_r$  completes the definition of each trajectory.

## **Appendix D: Force component derivations**

#### 645 D1 Derivation of Lift Direction Vector $e_L$

This appendix presents the derivation of the unit lift vector  $\mathbf{e}_L$ , expressed in the C-frame. The lift vector is orthogonal to the apparent wind velocity  $\mathbf{v}_a$ , and its orientation within the plane normal to  $\mathbf{v}_a$  is determined by the aerodynamic roll angle  $\phi_a$ . Since drag is aligned with the apparent wind direction by definition, the drag unit vector is

$$\mathbf{e}_D = \frac{\mathbf{v}_a}{\|\mathbf{v}_a\|} = \frac{1}{v_a} \begin{bmatrix} v_{\mathbf{a},\chi} \\ v_{\mathbf{a},\mathbf{n}} \\ v_{\mathbf{a},\mathbf{r}} \end{bmatrix}. \tag{D1}$$

To define  $\mathbf{e}_L$ , we first identify a basis for the plane orthogonal to  $\mathbf{v}_a$ . This is achieved by constructing a rotated frame  $\mathcal{A}$  whose  $\mathbf{e}_{\chi'}$ -axis is aligned with  $-\mathbf{v}_a$ . The  $(\mathbf{e}_{n'}, \mathbf{e}_{r'})$ -plane is then orthogonal to the wind vector.

The transformation from the C-frame to the A-frame consists of a rotation by  $-\chi_a$  around  $\mathbf{e}_r$  (aerodynamic heading), followed by a rotation by  $\gamma_a$  around the intermediate  $\mathbf{e}_{n'}$  axis (aerodynamic flight path angle).

The transformation matrix is

$$T_{\mathcal{A}\leftarrow C} = \begin{bmatrix} \cos\gamma_a\cos\chi_a & -\cos\gamma_a\sin\chi_a & \sin\gamma_a \\ \sin\chi_a & \cos\chi_a & 0 \\ -\sin\gamma_a\cos\chi_a & \sin\gamma_a\sin\chi_a & \cos\gamma_a \end{bmatrix}. \tag{D2}$$

Expressing v<sub>a</sub> in both reference frames yields

$$\begin{bmatrix} v_{a} \\ 0 \\ 0 \end{bmatrix} = \begin{bmatrix} \cos \gamma_{a} \cos \chi_{a} & -\cos \gamma_{a} \sin \chi_{a} & \sin \gamma_{a} \\ \sin \chi_{a} & \cos \chi_{a} & 0 \\ -\sin \gamma_{a} \cos \chi_{a} & \sin \gamma_{a} \sin \chi_{a} & \cos \gamma_{a} \end{bmatrix} \begin{bmatrix} v_{a,\tau} \\ v_{a,n} \\ v_{a,r} \end{bmatrix}$$
(D3)

Solving for the aerodynamic heading  $\chi_a$  and aerodynamic pitch  $\gamma_a$  from the radial and normal axis in Eq. (D3) we obtain

$$\tan \chi_a = -\frac{v_{a,n}}{v_{a,\chi}},\tag{D4}$$

$$\tan \gamma_a = \frac{v_{a,r}}{v_{a,\tau}}.$$
 (D5)

The unit vectors  $\mathbf{e}_{n'}$  and  $\mathbf{e}_{r'}$ , which span the plane perpendicular to  $\mathbf{v}_a$ , are found by applying the transformation matrix and simplifying

$$\mathbf{e}_{n'} = \frac{1}{v_{a,\tau}} \begin{bmatrix} -v_{a,n} \\ v_{a,\chi} \\ 0 \end{bmatrix},\tag{D6}$$

$$\mathbf{e}_{r'} = \frac{1}{v_a v_{a,\tau}} \begin{bmatrix} -v_{a,\chi} v_{a,r} \\ -v_{a,n} v_{a,r} \\ v_{a,\tau}^2 \end{bmatrix} . \tag{D7}$$

The aerodynamic roll angle  $\phi_a$  defines the orientation of  $\mathbf{e}_L$  within the  $(\mathbf{e}_{n'}, \mathbf{e}_{r'})$ -plane. By definition,  $\phi_a = 0$  corresponds to lift aligned with  $\mathbf{e}_{r'}$ , and positive  $\phi_a$  induces a clockwise rotation (right-hand turn) from the kite's perspective.

The lift direction is thus given by

$$\mathbf{e}_L = -\sin\phi_a \,\mathbf{e}_{n'} + \cos\phi_a \,\mathbf{e}_{r'}. \tag{D8}$$

Substituting the expressions for  $e_{n'}$  and  $e_{r'}$ , we obtain

$$\mathbf{e}_{L} = \frac{1}{v_{a}v_{a,\tau}} \begin{bmatrix} v_{a}v_{a,n}\sin\phi_{a} - v_{a,\chi}v_{a,r}\cos\phi_{a} \\ -v_{a}v_{a,\chi}\sin\phi_{a} - v_{a,n}v_{a,r}\cos\phi_{a} \\ v_{a,\tau}^{2}\cos\phi_{a} \end{bmatrix}.$$
 (D9)

This is the final expression for the lift direction vector in the C-frame, used in the main formulation of the aerodynamic force in Sect. 3.3.2.

# D2 Derivation of Angle of Attack $\alpha_{\rm w}$

We assume the kite remains aligned with the apparent wind  $\mathbf{v}_a$ . The angle of attack  $\alpha_{\rm w}$  is then obtained from the pitch angle between the total force at the bridle point  $\mathbf{F}_{\rm b}$  and the aerodynamic symmetry plane, corrected by the constant geometric pitch offset  $\theta_b$ .

Let  $\Pi_n$  denote the plane orthogonal to  $\mathbf{e}_{n'}$  (i.e. spanned by  $\mathbf{e}_{\chi}$  and  $\mathbf{e}_r$ ). The bridle force is projected onto  $\Pi_n$  as

$$\mathbf{F}_{\text{proj}} = \mathbf{F}_{\text{b}} - (\mathbf{F}_{\text{b}} \cdot \mathbf{e}_n) \, \mathbf{e}_n. \tag{D10}$$

The in-plane orientation of  $\mathbf{F}_{\text{proj}}$  defines the resultant force angle of attack  $\alpha_{\text{f}}$ ; consistent with the C-frame component ordering ( $\chi, n, r$ ), we write

$$\alpha_{\rm b} = \operatorname{atan2}(\mathbf{F}_{\rm proj} \cdot \mathbf{e}_{\chi}', -\mathbf{F}_{\rm proj} \cdot \mathbf{e}_{r}'). \tag{D11}$$

Finally, the effective angle of attack follows as

$$\alpha_{\rm w} = \alpha_{\rm b} - \theta_{\rm b}.$$
 (D12)

#### **D3** Derivation of Tether Force Components

This appendix provides the full derivation of the tether force components acting at the kite, based on a moment balance about the ground station. Two models for tether drag are considered: a distributed drag model and a simplified lumped approximation.

The tether is assumed to be straight and inertia-free, and only carries axial load. The net moment about the ground station must vanish

$$\mathbf{0} = \mathbf{r}_k \times \mathbf{F}_k + \mathbf{M}_q + \mathbf{M}_D, \tag{D13}$$

where

$$\mathbf{F}_k = -\mathbf{F}_t, \tag{D14}$$

$$\Rightarrow \mathbf{r}_k \times \mathbf{F}_t = \mathbf{M}_q + \mathbf{M}_D. \tag{D15}$$

Let  $\rho_t$  be the linear mass density of the tether. The differential gravitational force acting on a tether segment of length dl is

$$d\mathbf{F}_{g} = -\rho_{t}g\,\mathbf{e}_{z}\,dl = -\rho_{t}g\begin{bmatrix} \cos\chi\cos\beta\\ \sin\chi\cos\beta\\ \sin\beta \end{bmatrix}dl. \tag{D16}$$

Taking the moment about the ground station and integrating along the tether length gives

$$\mathbf{M}_g = \int_0^r \mathbf{r}(l) \times d\mathbf{F}_g, \quad \text{with } \mathbf{r}(l) = l\mathbf{e}_r, \tag{D17}$$

$$= \int_{0}^{r} \begin{bmatrix} 0 \\ 0 \\ l \end{bmatrix} \times d\mathbf{F}_{g},\tag{D18}$$

$$= \frac{\rho_t g r^2}{2} \begin{bmatrix} \sin \chi \cos \beta \\ -\cos \chi \cos \beta \\ 0 \end{bmatrix}. \tag{D19}$$

Assuming the total tether drag acts as a point force at the kite in the direction of the apparent wind (Vlugt et al., 2019), the lumped drag force becomes

$$\mathbf{D}_t = \frac{1}{8} \rho d_t r C_{D,c} v_a \mathbf{v}_a. \tag{D20}$$

The resulting moment is

$$\mathbf{M}_{D} = \mathbf{r}_{k} \times \mathbf{D}_{t} = \frac{1}{8} \rho d_{t} r^{2} C_{D,c} v_{a} \begin{bmatrix} -v_{w,n} \\ v_{w,\chi} - v_{\tau} \\ 0 \end{bmatrix}. \tag{D21}$$

https://doi.org/10.5194/wes-2025-205 Preprint. Discussion started: 17 October 2025 © Author(s) 2025. CC BY 4.0 License.

Inserting  $M_q$  and  $M_D$  into the moment balance and solving for  $F_t$ , the components of the tether force at the kite are

$$\mathbf{F}_{t} = \begin{bmatrix} F_{t,\tau} \\ F_{t,n} \\ F_{t,r} \end{bmatrix} = -\rho_{t}gr \begin{bmatrix} \frac{1}{2}\cos\chi\cos\beta \\ \frac{1}{2}\sin\chi\cos\beta \\ \sin\beta \end{bmatrix} + \frac{1}{8}\rho C_{D,c}d_{t}rv_{a} \begin{bmatrix} v_{w,\chi} - v_{\tau} \\ v_{w,n} \\ v_{w,r} - v_{\tau} \end{bmatrix} + \begin{bmatrix} 0 \\ 0 \\ -F_{tg} \end{bmatrix}.$$
 (D22)

If the tether behaves as a linear elastic spring, the radial ground station force is given by

$$F_{tg,r} = k_t(r - l_t), \tag{D23}$$

where  $k_t$  is the tether stiffness and  $l_t$  the unstretched tether length.

Author contributions. Conceptualisation, O.C., V.C. and R.S.; methodology, O.C. and V.C.; software, O.C; investigation, O.C. and V.C.; writing—original draft preparation, O.C.; writing—review and editing, O.C., V.C. and R.S.; supervision, R.S.; funding acquisition, R.S. All authors have read and agreed to the published version of the manuscript.

Competing interests. At least one of the (co-)authors is a member of the editorial board of Wind Energy Science. R.S. is a co-founder of and advisor for the start-up company Kitepower B.V., which is commercially developing a 100 kW kite power system and provided their test data used in this paper for validation. Both authors were financially supported by the European Union's MERIDIONAL project, which also provided funding for Kitepower B.V..

Acknowledgements. This work has been supported by the MERIDIONAL project, which receives funding from the European Union's Horizon Europe Programme under the grant agreement No. 101084216. The authors also gratefully acknowledge Kitepower for providing valuable validation data. We also acknowledge the use of OpenAI's ChatGPT and Grammarly for assistance in refining the writing style of this manuscript.

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

© Author(s) 2025. CC BY 4.0 License.



- Fechner, U., Vlugt, R. v. d., Schreuder, E., and Schmehl, R.: Dynamic model of a pumping kite power system, Renewable Energy, 83, 705–716, https://doi.org/10.1016/j.renene.2015.04.028, 2015.
- Hummel, J., Pollack, T., Eijkelhof, D., Van Kampen, E.-J., and Schmehl, R.: Power smoothing by kite tether force control for megawatt-scale airborne wind energy systems, Journal of Physics: Conference Series, 2767, 072 019, https://doi.org/10.1088/1742-6596/2767/7/072019, 2024.
- Hur, G.-B.: Identification of powered parafoil-vehicle dynamics from modelling and flight test data, Ph.D. thesis, Texas A&M University, 2005.
- Loyd, M. L.: Crosswind kite power (for large-scale wind power production), Journal of Energy, 4, 106–111, https://doi.org/10.2514/3.48021, publisher: American Institute of Aeronautics and Astronautics, 1980.
- Luchsinger, R. H.: Pumping Cycle Kite Power, in: Airborne Wind Energy, edited by Ahrens, U., Diehl, M., and Schmehl, R., pp. 47–64, Springer, Berlin, Heidelberg, ISBN 978-3-642-39965-7, https://doi.org/10.1007/978-3-642-39965-7, 3, 2013.
  - Malz, E., Koenemann, J., Sieberling, S., and Gros, S.: A reference model for airborne wind energy systems for optimization and control, Renewable Energy, 140, 1004–1011, https://doi.org/10.1016/j.renene.2019.03.111, 2019.
- Oehler, J. and Schmehl, R.: Aerodynamic characterization of a soft kite by in situ flow measurement, Wind Energy Science, 4, 1–21, https://doi.org/10.5194/wes-4-1-2019, 2019.
  - Oehler, J., Marc, v. R., and Schmehl, R.: Experimental investigation of soft kite performance during turning maneuvers, Journal of Physics: Conference Series, 1037, 052 004, https://doi.org/10.1088/1742-6596/1037/5/052004, 2018.
  - Paulig, X., Bungart, M., and Specht, B.: Conceptual Design of Textile Kites Considering Overall System Performance, in: Airborne Wind Energy, edited by Ahrens, U., Diehl, M., and Schmehl, R., pp. 547–562, Springer, Berlin, Heidelberg, ISBN 978-3-642-39965-7, https://doi.org/10.1007/978-3-642-39965-7\_32, 2013.
  - Poland, J. and Schmehl, R.: Modelling Aero-Structural Deformation of Flexible Membrane Kites, Energies, 16, 5264, https://doi.org/10.3390/en16145264, 2023.
  - Poland, J. and Schmehl, R.: TUDELFT\_V3\_KITE, https://github.com/awegroup/TUDELFT\_V3\_LEI\_KITE, 2024a.
- Poland, J. A. W. and Schmehl, R.: A virtual wind tunnel for deforming airborne wind energy kites, Journal of Physics: Conference Series, 2767, 072 001, https://doi.org/10.1088/1742-6596/2767/7/072001, 2024b.
  - Poland, J. A. W., van Spronsen, J. M., Gaunaa, M., and Schmehl, R.: Wind Tunnel Load Measurements of a Leading-Edge Inflatable Kite Rigid Scale Model, Wind Energy Science Discussions, pp. 1–33, https://doi.org/10.5194/wes-2025-77, publisher: Copernicus GmbH, 2025.
- Ranneberg, M., Wölfle, D., Bormann, A., Rohde, P., Breipohl, F., and Bastigkeit, I.: Fast Power Curve and Yield Estimation of Pumping
  Airborne Wind Energy Systems, in: Green Energy and Technology, pp. 623–641, ISBN 978-981-10-1946-3, https://doi.org/10.1007/978-981-10-1947-0\_25, journal Abbreviation: Green Energy and Technology, 2018.
  - Schelbergen, M.: Power to the airborne wind energy performance model, Ph.D. dissertation, Delft University of Technology, https://resolver.tudelft.nl/uuid:353d390a-9b79-44f1-9847-136a6b880e12, 2024.
- Schelbergen, M. and Schmehl, R.: Validation of the quasi-steady performance model for pumping airborne wind energy systems, Journal of Physics: Conference Series, 1618, 032 003, https://doi.org/10.1088/1742-6596/1618/3/032003, publisher: IOP Publishing, 2020.
  - Schmehl, R., Noom, M., and van der Vlugt, R.: Traction Power Generation with Tethered Wings, in: Airborne Wind Energy, edited by Ahrens, U., Diehl, M., and Schmehl, R., pp. 23–45, Springer, Berlin, Heidelberg, ISBN 978-3-642-39965-7, https://doi.org/10.1007/978-3-642-39965-7\_2, 2013.

https://doi.org/10.5194/wes-2025-205 Preprint. Discussion started: 17 October 2025

© Author(s) 2025. CC BY 4.0 License.

- Schmidt, W. and Anderson, W.: Kites: Pioneers of Atmospheric Research, in: Airborne Wind Energy, edited by Ahrens, U., Diehl, M., and Schmehl, R., pp. 95–116, Springer, Berlin, Heidelberg, ISBN 978-3-642-39965-7, https://doi.org/10.1007/978-3-642-39965-7\_6, 2013.
  - Sánchez-Arriaga, G., García-Villalba, M., and Schmehl, R.: Modeling and dynamics of a two-line kite, Applied Mathematical Modelling, 47, 473–486, https://doi.org/10.1016/j.apm.2017.03.030, 2017.
  - Terink, E., Breukels, J., Schmehl, R., and Ockels, W.: Flight Dynamics and Stability of a Tethered Inflatable Kiteplane, Journal of Aircraft, 48, 503–513, https://doi.org/10.2514/1.C031108, 2011.
- 800 Thedens, P. and Schmehl, R.: An Aero-Structural Model for Ram-Air Kite Simulations, Energies, 16, 2603, https://doi.org/10.3390/en16062603, 2023.
  - van Deursen, V.: Dynamic Simulation Techniques for Airborne Wind Energy Systems, Master's dissertation, TU Delft, https://repository.tudelft.nl/record/uuid:bb32fc5b-300a-4789-9b48-90927f035378, 2024.
- Vermillion, C., Cobb, M., Fagiano, L., Leuthold, R., Diehl, M., Smith, R. S., Wood, T. A., Rapp, S., Schmehl, R., Olinger, D., and Demetriou,
  M.: Electricity in the air: Insights from two decades of advanced control research and experimental flight testing of airborne wind energy systems, Annual Reviews in Control, 52, 330–357, https://doi.org/10.1016/j.arcontrol.2021.03.002, 2021.
  - Viré, A., Lebesque, G., Folkersma, M., and Schmehl, R.: Effect of Chordwise Struts and Misaligned Flow on the Aerodynamic Performance of a Leading-Edge Inflatable Wing, Energies, 15, https://doi.org/10.3390/en15041450, 2022.
- Vlugt, R. v. d., Peschel, J., and Schmehl, R.: Design and Experimental Characterization of a Pumping Kite Power System, in: Air810 borne Wind Energy, edited by Ahrens, U., Diehl, M., and Schmehl, R., pp. 403–425, Springer Berlin Heidelberg, Berlin, Heidelberg, https://doi.org/10.1007/978-3-642-39965-7\_23, 2013.
  - Vlugt, R. v. d., Bley, A., Noom, M., and Schmehl, R.: Quasi-steady model of a pumping kite power system, Renewable Energy, 131, 83–99, https://doi.org/10.1016/j.renene.2018.07.023, 2019.