# Peer review of "Translational Dynamics of Bridled Kites: A Reduced-Order Model in the Course Reference Frame"

_Wind Energy Science, 2025_

## Author Comment (AC1)

**Response to Reviewer Comments**

*Manuscript title: "Translational Dynamics of Bridled Kites: A Reduced-Order Model in the Course Reference Frame"*

We thank the Reviewer for the positive assessment of our work and the helpful comments. Below we provide a point-by-point response. Reviewer comments appear in *italics*, followed by our responses.

**Minor Comments**

*\* When $\theta_b$ is introduced on line 94, the formula is not yet clear, that only comes on line 104. I recommend making a forward reference to Equation (2) and adding $\theta_b$ in Figure 2.*

We thank the reviewer for this suggestion. In the revised manuscript, we now add a forward reference to Eq. (2) at the point where $\theta_b$ is first introduced. We also updated Figure 2 to include the angle $\theta_b$, ensuring visual consistency with the equations and narrative. This improves clarity for the reader.

*\* All values in Equation (3) should be defined below it.*

We agree with the reviewer. We have updated the text following Eq. (3) to define all variables used in the expression, including the yaw rate $\dot{\psi}$ and apparent wind speed $v_a$. This ensures the equation is self-contained and understandable to all readers.

*\* On line 174, "equations of obtained" should be "equations obtained"*

Thank you for spotting this typo. We have corrected the phrase to "equations obtained" in the revised manuscript.

*\* The radial velocity is also set to zero "for practical implementation" on line 296. I understand that this is not used as equilibrium condition, but it is strange to mention this so late in the process, after emphasizing that only the tangential acceleration is set to zero.*

We thank the reviewer for this observation. In the revised manuscript, we clarify that the radial acceleration $\dot{v}_r$ is treated as a user-controlled input—both in the dynamic and quasi-steady models—and is not part of the equilibrium condition. The quasi-steady formulation is strictly defined by the vanishing of tangential acceleration, $\dot{v}_\tau = 0$. While the radial motion can be held constant in some simulations for simplicity, this is not an inherent part of the quasi-steady model definition, and we have removed the earlier statement to avoid confusion.

*\* On line 383, the gaps in the curves of Figure 8 are explained, but this could be explained in more detail. What is observed if the model is applied to these situations?*

We thank the reviewer for this helpful comment. The gaps in the estimated quantities correspond to situations where the model cannot resolve a quasi-steady solution—i.e., no tangential speed (or, equivalently, angle of attack) can be found that satisfies the force equilibrium given the measured state. This is analogous to the absence of a zero crossing in the tangential acceleration curve shown in Fig. 6.

This behaviour is particularly evident in the TU Delft V3 dataset, where the system was equipped with a disproportionately large KCU and a relatively heavy tether compared to its small wing area. As a result, certain flight segments, especially during the climbing phase after a turn, fall outside the validity domain of the quasi-steady model. In these cases, applying the model yields no physical solution, and the corresponding points are omitted in the plots to reflect this.

We have revised the text in Subsection 5.2 to clarify this explanation and highlight the conditions under which the model fails to find a solution.

*\* Missing text on line 487.*

We thank the reviewer for catching this. The affected subsection has been revised, and the missing sentence is now complete.

---

## Author Comment (AC2)

**Response to Reviewer Comments**

*Manuscript title: "Translational Dynamics of Bridled Kites: A Reduced-Order Model in the Course Reference Frame"*

We thank the Reviewer for the thoughtful comments and suggestions. Below we provide a point-by-point response. Reviewer comments appear in *italics*, followed by our responses.

**General Comments**

*The two major comments/corrections required are the following:*

*1. Subsection 2.1 should be revised to improve clarity. As outlined in the specific comments, the authors are expected to clarify the definitions of the variables, the figure, and the equations. Enhancing the flow and depth of this subsection will improve the understanding of the physical principles.*

We thank the reviewer for the suggestion. In the revised manuscript, the previous Subsections 2.1 and 2.2 have been consolidated into a single subsection to improve clarity and narrative flow. Within this unified section, we have clarified the definitions of key variables (e.g., the tow angle $\lambda_b$ and geometric pitch offset $\theta_b$, improved figure placement and referencing, and expanded the explanation of the trim condition and depower actuation. These revisions enhance the readability of the section and strengthen the link between the physical interpretation and the modelling assumptions.

*2. Subsection 6.3: comparison between soft and rigid wings lacks clarity. Please add a clear description of the rigid-wing principles and variables/equations needed for the comparison, or remove the section.*

We thank the reviewer for this observation. In the revised manuscript, we have clarified the modelling assumptions for the rigid-wing simulations at the start of Subsection 6.3. Specifically, we now state that the rigid-wing kite (based on the Ampyx AP2) is assumed to maintain a constant orientation relative to the resultant tether force, which allows prescribing a fixed pitch angle offset without explicitly modelling the rotational dynamics. We have added a detailed explanation of how the pitch angle and angle of attack are set based on this assumption, and we refer the reader to Appendix B for the full aerodynamic and mass properties. These updates aim to improve clarity for readers who may not be familiar with rigid-wing AWES modeling conventions.

**Specific Comments**

**Subsection 2.1: Longitudinal static stability and trim condition**

*Figure 2: Neither the caption nor the text describe all the variables in the drawing. You do not mention $\theta_k$, $r$, or $F_b$; moreover $\theta_b$ should be in the figure. Also, it is not explained why the front chord line is perpendicular to the chord line when $\theta_d = 0$. Please adjust the figure and caption coherently.*

We thank the reviewer for this helpful observation. To improve clarity, we revised the caption of Figure 2 so that all variables appearing in the schematic are explicitly defined and consistent with the notation used throughout the manuscript. The caption now describes (i) the resultant bridle force $\vec{F}_b$, (ii) the position vector $\vec{r}$ from the ground station to the kite, (iii) the pitch angle

$\theta_k$ between $\vec{F}_b$ and $\vec{r}$, and (iv) the geometric pitch angle $\theta_b$, which relates the bridle and wing angles of attack through Eqs. (1)–(2).

We also added a brief explanation stating that the configuration with $\theta_d = 0°$ is used as a reference trim state in which the front bridle line is perpendicular to the local chord line, a choice consistent with typical powered trim geometries of LEI kites, while acknowledging that the exact value may vary between designs. These clarifications improve the consistency and readability of the schematic.

**Subsection 4.1: Definition and assumptions**

*Figure 6 caption: Which angle of attack is used for the left image? Which wind velocity? Which velocity is used for the right plot? The acceleration scale is extremely large (hundreds of g). The difference between the curves should be 9.81 m/s$^2$, but appears to be ∼1000 m/s$^2$. Please clarify.*

Thank you for this careful observation. You are absolutely right—the originally submitted plot mistakenly showed the *tangential force* rather than the *tangential acceleration*, which caused the apparent magnitude issue (orders of magnitude above $9.81 \, \mathrm{m\,s^{-2}}$). This has now been corrected in the revised figure, and the acceleration range is consistent with the expected values.

The angle of attack plotted corresponds to the instantaneous value associated with each tangential speed, resulting from the geometric relations detailed in the model.

To improve clarity and better illustrate the equilibrium points, we have also revised the simulation conditions by lowering the wind speed to $6 \, \mathrm{m\,s^{-1}}$, which makes the two equilibrium crossings more clearly visible. These two crossings emerge due to the nonlinear behaviour of the aerodynamic polars: there can be two angles of attack resulting in zero net acceleration. However, only one of these corresponds to a physically feasible and locally stable operating point.

The figure caption has been updated accordingly to reflect these clarifications, and also a small clarification is added to the text.

**Subsection 6.3: Dynamic response in crosswind trajectories**

*Line 515–518: In the rigid-wing case, how do you compute the pitch angle and the angle of attack? How do you choose the trajectory radius? This influences the results. Consider referring to Fig. 10 of "Trimming a fixed-wing AWES for coordinated circular flights" (doi:10.5194/wes-2025-193).*

We thank the reviewer for this insightful comment. In the rigid-wing simulations we do not compute the pitch angle dynamically. Instead, to enable a consistent comparison with the soft-kite framework, we prescribe a fixed angle between the wing chord and the tether force direction. This is a valid approximation for bridled soft-kite systems—where the bridle passively maintains alignment with the tether force—but represents a simplification in the rigid-wing case. The chosen pitch offset yields a trim angle of attack of approximately 6° for a massless kite, consistent with measured flight values for the AP2 system.

We agree that the curvature of the trajectory influences the equilibrium conditions. To assess this effect, we simulate two circular paths with different turning radii, holding the reeling speed constant. This allows us to explore how curvature affects the inertial loads and deviations from quasi-steady behaviour.

To help contextualise our assumptions, we now reference the coordinated-flight trim analysis of (doi:10.5194/wes-2025-193).We clarify in the manuscript that our approach deliberately simplifies this coupling in order to focus on the influence of wing loading within a unified modelling framework.

*Line 520: You could strengthen your conclusion by comparing with the Loyd-mode approximation (eq. 85 or 87) in Trevisi et al., "Flight stability of rigid-wing AWES" (doi:10.3390/en14227704).*

We thank the reviewer for highlighting this valuable connection. We have now included an explicit comparison with the Loyd-mode approximation from (doi:10.3390/en14227704), using the expression provided in their Eq. (85), which linearizes the system around the quasi-steady solution. In the revised manuscript (see Table 1 and corresponding discussion), we compute the dominant eigenvalue $\lambda_{\text{Loyd}}$ for each kite configuration based on mean values extracted from the dynamic simulations. This eigenvalue captures the strength of the restoring dynamics and reflects the system's tendency to return to the quasi-steady state.

As expected, the results show a clear trend: soft kites, with lower wing loading, exhibit significantly more negative values of $\lambda_{\text{Loyd}}$ (around $-4$ to $-5\,\text{s}^{-1}$), indicating faster convergence. In contrast, rigid wings yield values closer to $-1\,\text{s}^{-1}$, consistent with their higher inertia and slower dynamic response. This supports our interpretation of the quasi-steady state as a moving attractor and demonstrates that the Loyd-mode eigenvalue effectively captures the role of wing loading in shaping the system's transient behaviour.

**Technical Corrections**

*Line 77: "about the bridle point (B)": Please introduce Figure 2 before referring to it.*

We thank the reviewer for this helpful suggestion. We have revised the text to introduce Figure 2 in the same sentence.

*Line 79: Are $x_{cp}$ and CP (in Fig. 2 and line 87) the same? If yes, please use the same notation.*

We thank the reviewer for pointing out this inconsistency. We are now using CP only.

*Line 94: "a constant geometric angle $\theta_b$": Is this $\theta_b$ or $\lambda_b$? $\theta_b$ is never introduced before; please clarify why it is approximated as constant.*

We thank the reviewer for pointing this out. As explained in the major comments, we restructured the section and now this should be clear.

*Line 102: Please explain the physical meaning and derivation of Equations (1) and (2) step by step.*

Thank you. The restructure of this section should make Eqs. (1) and (2) clear now.

*Line 137: Please define the variables in Eq. (3), especially yaw rate $\dot{\psi}$ and apparent velocity $v_a$, which is shown in Fig. 2 but never defined.*

We thank the reviewer for pointing this out. We have revised the text to explicitly define all variables in Eq. (3), including the yaw rate $\dot{\psi}$ and the apparent wind speed $v_a$, for clarity.

*Figure 5 and line 255: Please describe in the caption the variables introduced for the first time. $F_{t,g}$ appears here but is never explicitly cited in the text. Also ensure consistent notation between $F_{t,g}$ and $F_{tg}$.*

We thank the reviewer for pointing this out. The figure caption has been revised to explicitly define $F_{t,g}$ as the tether force at the ground and $F_k$ as the resultant force at the kite. We also clarified the meaning of the distributed forces shown in the diagram: the orange distribution

represents tether weight, and the blue distribution represents tether drag. We have also ensured consistent use of the subscript notation $F_{t,g}$ throughout.

*Line 240: Only the perpendicular component of apparent velocity generates aerodynamic drag on the tether. Equations (24–26) seem not to reflect this, and may overestimate drag.*

We thank the reviewer for this accurate observation. In the original manuscript, the drag expressions were obtained without projecting the apparent wind velocity onto the plane normal to the (straight) tether. We have revised Eqs. (24)–(26) such that the aerodynamic drag is computed using only the components of the apparent velocity perpendicular to the tether. All affected plots have been updated accordingly.

*Line 270: "crosses zero at two points": In Figure 6 only one point is shown. Please clarify.*

We thank the reviewer for pointing this out. In the original figure, the second zero-crossing was not visible due to the choice of wind speed. We have reduced the wind speed in the revised figure to reveal both equilibrium points. As discussed in the text, one crossing corresponds to a physically feasible angle of attack near the nominal operating point, while the second occurs at a high angle of attack well beyond stall and is not aerodynamically viable.

*Lines 320–322: The control input $u_p$ is not mentioned here. Was this intentional?*

We thank the reviewer for catching this omission. The absence of the control input $u_p$ in this part of the text was unintentional. We have updated the relevant sentence to explicitly include $u_p$.

*Line 347: "a validated lifting-line-based model": Previously you state that lifting-line and CFD struggle to capture the aerodynamics of these systems. Please clarify how the validation was performed in the cited works.*

We appreciate this important clarification request. In the first mention (Line 271), we have removed the term "validated" and instead described the Vortex Step Method (VSM) as a lifting-line model tailored to soft kites. In the second instance (Line 347), we retain the validation claim and have clarified in the revised text that the model was recently validated against wind tunnel measurements as reported in (Poland, et.al. 2025).

*Line 364: Why cite an "onboard turbine" here? Are we referring to soft kites?*

We thank the reviewer for this comment. The onboard turbine referenced here is indeed specific to soft kite systems and is typically used to power the electronics of the kite control unit (KCU). We have clarified this in the revised text.

*Line 412: "assumption of straight flight": Do you mean straight upward?*

We thank the reviewer for this question. By "straight flight," we refer to a trajectory with zero course rate—that is, the kite maintains a constant course angle. We removed the term straight flight and substitued by a constant course angle to avoid confusion.

*Line 436, Eq. (44): Should the factor be $\frac{1}{2}$ instead of 2? Also, please add a derivation. Should $v_{r,\mathrm{opt}}$ be $v_w \cos\phi \cos\beta - v_{r,\mathrm{opt}}$?*

We thank the reviewer for this comment. The original expression is correct: the factor of 2 in Eq. (44) arises after substituting the expression for the optimal apparent wind speed into the aerodynamic force formula, which already includes the standard $\frac{1}{2}$ dynamic pressure term. To clarify this, we have added a concise derivation based on triangle similarity, consistent with Schmehl et. al. (2013), and made the substitution step explicit in the text.

*Figure 10: The reeling factor is mentioned but never defined. Please define it.*

We thank the reviewer for this comment. The reeling factor is now clearly defined in the caption

of Fig. 10 as the ratio $f = v_r/v_w$ between reeling speed and wind speed.

*Line 467: Please cite explicitly the two types of $\Delta_\phi$ used (circular vs figure-eight).*

We thank the reviewer for this suggestion. We have updated the text to explicitly cite and distinguish the two values of $\Delta_\phi$ used for circular and figure-eight trajectories, respectively.

*Line 487: Missing sentence.*

We thank the reviewer for pointing this out. The entire subsection has been revised, and all sentences are now complete.

*Line 542: "As the kite moves with": Do you mean "As the kite moves in"?*

We thank the reviewer for spotting this. We have corrected the phrase to "as the kite moves in" in the revised manuscript.